# Towards Explaining Distribution Shift

## Abstract

A distribution shift can have fundamental consequences such as signaling a change in the operating environment or significantly reducing the accuracy of downstream models. Thus, understanding distribution shifts is critical for examining and hopefully mitigating the effect of such a shift. Most prior work has focused on merely detecting if a shift has occurred and assumes any detected shift can be understood and handled appropriately by a human operator. We hope to aid in these manual mitigation tasks by explaining the distribution shift using interpretable transportation maps from the original distribution to the shifted one. We derive our interpretable mappings from a relaxation of the optimal transport problem, where the candidate mappings are restricted to a set of interpretable mappings. We then use quintessential examples of distribution shift in simulated and real-world cases to showcase how our explanatory mappings provide a better balance between detail and interpretability than the de facto standard mean shift explanation by both visual inspection and our PercentExplained metric.

## 1 Introduction

Most real-world environments are constantly changing, and in many situations, understanding how a specific operating environment has changed is crucial to making decisions respective to such a change. Such a change might be a new data distribution seen in deployment which causes a machine learning model to begin to fail. Another example is a decrease in monthly sales data which could be due to a temporary supply chain issue in distributing a product or could mark a shift in consumer preferences. When these changes are encountered, the burden is often placed on a human operator to investigate the shift and determine the appropriate reaction, if any, that needs to be taken. In this work, our goal is to aid these operators in providing an explanation of such a change.

This ubiquitous phenomenon of having a difference between related distributions is known as distribution shift. Much prior work focuses on *detecting* distribution shifts; however, there is little prior work that looks into *understanding* a detected distribution shift. As it is usually solely up to an operator investigating a flagged distribution shift to decide what to do next, understanding the shift is critical for the operator to more efficiently mitigate any harmful effects of the distribution shift. Without a defined approach to this task, the current de facto standard in analyzing a shift is looking at how the mean of the original, *source*, distribution shifted to the new, *target*, distribution. However, this simple explanation can miss crucial shift information due to being a coarse summary (e.g., Fig. 2). Further, in high-dimensional regimes, a shift in means could be uninterpretable due to its high dimensionality. Instead, if after flagging that a shift has occurred, we could automatically provide more detailed information about the shift but still remain at a level that is interpretable, we could reduce the manual load on the operator to understand the shift, and, ultimately, to take action if necessary.

Therefore, we propose a novel framework for explaining distribution shifts, such as showing how features have changed or how groups within the distributions have shifted. Since a distribution shift can be seen as a movement from a source distribution $\boldsymbol{x} \sim P_{src}$ to a target distribution $\boldsymbol{y} \sim P_{tgt}$, we define a distribution shift explanation as a transport map $T(\boldsymbol{x})$ which maps a point in our source distribution onto a point in our target distribution. For example, under this framework, the typical distribution shift explanation via mean shift can be written as $T(\boldsymbol{x}) = \boldsymbol{x} + (\mu_{\boldsymbol{y}} - \mu_{\boldsymbol{x}})$. Intuitively, these transport maps can be thought of as a functional approximation of how the source distribution could have moved in a distribution space to become our target distribution. However, without making assumptions on the type of shift, there exist many possible mappings that explain the shift (see subsection A.2 for examples). Thus, we leverage prior optimal transport work to define an

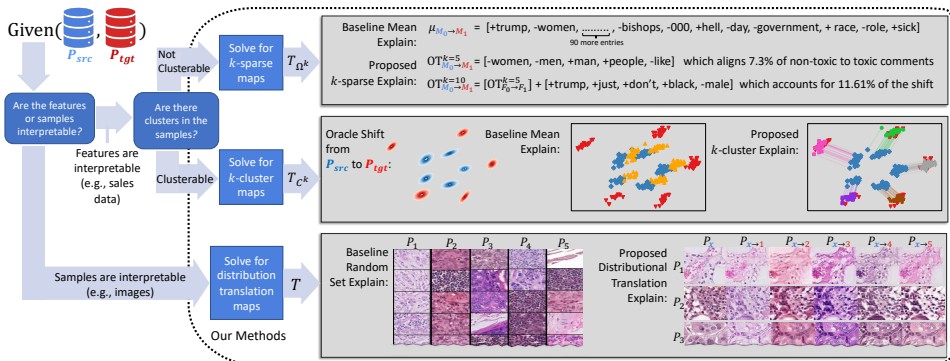

Figure 1: An overall look at our approach to explaining distribution shifts, where given a source $P_{src}$ and shifted $P_{tgt}$ dataset pair, a user can choose to explain the distribution shift using $k$-sparse maps (which are best suited for high dimensional or feature-wise complex data), $k$-cluster maps (for tracking how heterogeneous groups change across the shift), or distribution translation maps (for data which has uninterpretable raw features such as images). For details on the results seen in the three boxes, please see experiments in Section 5 and Section 6.

ideal distribution shift explanation and develop practical algorithms for finding and validating such maps. We summarize our contributions as follows:

- In Section 3, we define interpretable transport maps by constraining a relaxed form of the optimal transport problem to only search over a set of interpretable mappings and suggest possible interpretable sets.

- In Section 4, we develop practical methods for finding such interpretable mappings which allow us to adjust the interpretability of an explanation to fit the needs of a situation.

- In Section 5, we show empirical results on real-world tabular datasets demonstrating how our explanations and our PercentExplained metric can aid an operator in understanding how a distribution has shifted.

- In Section 6, we use latent transport mappings and Image-to-Image translation methods to extend this approach to explain image-based shifts such as investigating how the staining of histopathological images varies across a hospital network.

## 2 RELATED WORKS

The characterization of the problem of distribution shift has been extensively studied (Quiñonero-Candela et al., 2009; Storkey, 2009; Moreno-Torres et al., 2012) via breaking down a joint distribution $P(\boldsymbol{x}, y)$ of features $\boldsymbol{x}$ and outputs $y$, into conditional factorizations such as $P(y|\boldsymbol{x})P(\boldsymbol{x})$ or $P(\boldsymbol{x}|y)P(y)$. For covariate shift (Sugiyama et al., 2007) the $P(\boldsymbol{x})$ marginal differs from source to target, but the output conditional $P(y|\boldsymbol{x})$ the same, while label shift (also known as prior probability shift) (Zhang et al., 2013; Lipton et al., 2018) is when the $P(y)$ marginals differ from source to target, but the full-feature conditional $P(\boldsymbol{x}|y)$ remains the same. In this work, we refer to general problem distribution shift, i.e. a shift in the joint distribution (with no distinction between $y$ and $\boldsymbol{x}$), and leave applications of explaining specific sub-genres of distribution shift to future work.

As far as we are aware, this is the first work specifically tackling explaining distribution shifts; however, there are distinct works that can be applied to explain distribution shifts. For example, one could use feature attribution methods Saarela & Jauhiainen (2021); Molnar (2020) on a domain/distribution classifier (e.g., Shanbhag et al. (2021) uses Shapley values Shapley (1997) to explain how changing input feature distributions affect a classifier's behavior), or once could find samples which are most illustrative of the differences between distributions Brockmeier et al. (2021). Additionally, one could use counterfactual generation methods Karras et al. (2019); Sauer & Geiger (2021); Pawelczyk et al. (2020) and apply them for "distributional counterfactuals" which would show what a sample from $P_{tgt}$ would have looked like if it instead came from $P_{src}$ (e.g., Pawelczyk et al. (2020) uses a classifier-guided VAE to generate class counterfactuals on tabular data). We explore this distributional counterfactual explanation approach in subsection 6.2.

A sister field is that of detecting distribution shifts. This is commonly done using methods such as statistical hypothesis testing of the input features (Nelson, 2003; Rabanser et al., 2018; Quiñonero-Candela et al., 2009), training a domain classifier to test between source and non-source domain samples Lipton et al. (2018), etc. In (Kulinski et al., 2020; Budhathoki et al., 2021), the authors

attempt to provide more information beyond the binary "has a shift occurred?" via localizing a shift to a subset of features or causal mechanisms. (Kulinski et al., 2020) does this by introducing the notion of Feature Shift, which first detects if a shift has occurred and if so, localizes that shift to a specific subset of features that have shifted from source to target. In (Budhathoki et al., 2021), the authors take a causal approach via individually factoring the source and target distributions into a product of their causal mechanisms (i.e. a variable conditioned on its parents) using a shared causal graph, which is assumed to be known/discoverable. Then, the authors "replace" a subset of causal mechanisms from $P_{src}$ with $P_{tgt}$, and measure divergence from $P_{src}$ (i.e. measuring how much the subset change affects the source distribution). Both of these methods are still focused on detecting distribution shifts (via identifying shifted causal mechanisms or feature-level shifts), unlike explanatory mappings which help explain *how* the data has shifted.

## 3 EXPLAINING DISTRIBUTION SHIFTS VIA TRANSPORT MAPS

The underlying assumption of distribution shift is that there exists a relationship between the source and target distributions. From a distributional standpoint, we can view distribution shift as a *movement*, or transportation, of samples from the source distribution $P_{src}$ to the target distribution $P_{tgt}$. Thus, we can capture this relationship between the distributions via a transport map $T$ from the source distribution to the target, i.e., if $\boldsymbol{x} \sim P_{src}$, then $T(\boldsymbol{x}) \sim P_{tgt}$. If this mapping is understandable to an operator investigating a distribution shift, this can serve as an explanation to the operator on what changed between the environments; thus allowing for more effective reactions to the shift. Therefore, we claim a shift explanation to be: *an interpretable transport map $T$ which approximately maps a source distribution $P_{src}$ onto a target distribution $P_{tgt}$ such that $T_\sharp P_{src} \approx P_{tgt}$.* Similar to ML model interpretability Molnar (2020), an interpretable map can either be one that is intrinsically interpretable (subsection 3.1) or a mapping which is explained via post-hoc methods such as sets of input-output pairs (subsection 6.2).

### 3.1 INTRINSICALLY INTERPRETABLE TRANSPORTATION MAPS

In order to find such a mapping between distributions, it is natural to look to Optimal Transport (OT) and its extensive prior work in this field Cuturi (2013); Arjovsky et al. (2017); Torres et al. (2021); Peyré & Cuturi (2019). An OT mapping is defined as a method of optimally moving points from one distribution to another given a transport cost function $c$ and is formally defined as:

$$T_{OT} := \arg\min_T \mathbb{E}_{P_{src}}\left[c(\boldsymbol{x}, T(\boldsymbol{x}))\right] \text{ s.t.} T_\sharp P_{src} = P_{tgt}$$

where $T_\sharp P_{src}$ is the push forward operator that can be viewed as applying the $T$ mapping onto all points in $P_{src}$, and $T_\sharp P_{src} = P_{tgt}$ is the marginal constraint, meaning that $T_{OT}$ must perfectly align the source distribution and the target.

OT is a natural starting point for shift explanations as it allows for a rich geometric structure on the space of distributions, and by finding a mapping that minimizes a transport cost we are forcing our mapping to retain as much information about the original $\boldsymbol{x}$ samples when aligning $P_{src}$ and $P_{tgt}$. For more details about OT, please see Villani (2009); Peyré & Cuturi (2019).

However, since OT considers all possible mappings which satisfy the marginal constraint, this means the resulting $T_{OT}$ can be arbitrarily complex and thus possibly uninterpretable as a shift explanation. We can alleviate this by restricting the candidate transport maps to belong to a set of user-defined interpretable mappings $\Omega$. However, this problem can be infeasible if $\Omega$ does not contain a mapping that satisfies the marginal constraint. Therefore, we can use Lagrangian relaxation to relax the marginal constraint, giving us an *Interpretable Transport* mapping $T_{IT}$:

$$T_{IT} := \arg\min_{T \in \Omega} \mathbb{E}_{P_{src}}\left[c(\boldsymbol{x}, T(\boldsymbol{x}))\right] + \lambda \, \phi(P_{\mathrm{T}(\boldsymbol{x})}, P_{tgt}) \tag{1}$$

where $c$ is assumed to be the squared Euclidean cost and $\phi(\cdot, \cdot)$, the divergence function, is assumed to be the squared Wasserstein-2 metric, unless stated otherwise. Due to the heavily complex and context-specific nature of distribution shift, it is likely that $\Omega$ would be initialized based on context. However, we suggest two *general* methods in the next section as a starting point and hope that future work can build upon this framework for specific contexts.

### 3.2 INTERPRETABLE TRANSPORT SETS

The current common practice for explaining distribution shifts is comparing the means of the source and the target distributions. The mean shift explanation can be generalized as $\Omega_{\text{vector}} = \{T : T(\boldsymbol{x}) =$

$\boldsymbol{x}+\delta\}$ where $\delta$ is a constant vector and mean shift being the specific case where $\delta$ is the difference of the source and target means. By letting $\delta$ be a function of $\boldsymbol{x}$, which further generalizes the notion of mean shift by allowing each point to move a variable amount per dimension, we arrive at a transport set that includes any possible mapping $T : \mathbb{R}^d \to \mathbb{R}^d$. However, even a simple transport set like $\Omega_{\text{vector}}$ can yield uninterpretable mappings in high dimensional regimes (e.g., a shift vector of over 100 dimensions). To combat this, we can regulate the complexity of a mapping by forcing it only move points along a specified number of dimensions. We define this as $k$-*Sparse Transport*:

$k$-**Sparse Transport:**    For a given class of transport maps, $\Omega$ and a given $k \in \{1, ..., d\}$, we can find a subset $\Omega_{sparse}^{(k)}$ which is the set of transport maps from $\Omega$ which only transport points along $k$ dimensions or less. Formally, we define an active set $\mathcal{A}$ to be the set of dimensions along which a given $T$ moves points: $\mathcal{A}(T) \triangleq \{j \in \{1, \ldots, d\} : \exists \boldsymbol{x}, T(\boldsymbol{x})_j - x_j \neq 0\}$. Then, we define $\Omega_{sparse}^{(k)} = \{T \in \Omega : |\mathcal{A}(T)| \leq k\}$.

$k$-sparse transport is most useful in situations where a distribution shift has happened along a subset of dimensions, such as explaining a shift where some sensors in a network are picking up a change in an environment. However, in situations where points shift in different directions based on their original value, e.g., when investigating how a heterogeneous population responded to an advertising campaign, $k$-sparse transport is not ideal. Thus, we provide a shift explanation that breaks the source and target distributions into $k$ sub-populations and provides a vector-based shift explanation per sub-population. We define this as $k-cluster\ transport$:

$k$-**Cluster Transport**    Given a $k \in \{1, \ldots, d\}$ we define $k$-cluster transport to be a mapping which moves each point $\boldsymbol{x}$ by constant vector which is specific to $\boldsymbol{x}$'s cluster. More formally, we define a labeling function $\sigma(\boldsymbol{x}; M) \triangleq \arg\min_j \|\boldsymbol{m}_j - \boldsymbol{x}\|_2$, which returns the index of the column in $M$ (i.e. the label of the cluster) which $\boldsymbol{x}$ is closest to. With this, we define $\Omega_{\text{cluster}}^{(k)} = \left\{T : T(\boldsymbol{x}) = \boldsymbol{x} + \delta_{\sigma(\boldsymbol{x};M)}, M \in \mathbb{R}^{d \times k}, \Delta \in \mathbb{R}^{d \times k}\right\}$, where $\delta_j$ is the $j^{\text{th}}$ column of $\Delta$.

Since measuring the exact interpretability of a mapping is heavily context-dependent, we can instead use $k$ in the above transport maps to define a partial ordering of interpretability of mappings *within* a class of transport maps. Let $k_1$ and $k_2$ be the size of the active sets for $k$-sparse maps (or the number of clusters for $k$-cluster maps) of $T_1$ and $T_2$ respectively. If $k_1 \leq k_2$, then $\text{Inter}(T_1) \geq \text{Inter}(T_2)$, where $\text{Inter}(T)$ is the interpretability of shift explanation $T$. For example, we claim the interpretability of a $T_1 \in \Omega_{sparse}^{(k=10)}$ is greater than (or possibly equal to) the interpretability of a $T_2 \in \Omega_{sparse}^{(k=100)}$ since a shift explanation in $\Omega$ which moves points along only 10 dimensions is more interpretable than a similar mapping which moves points along 100 dimensions. A similar result can be shown for $k$-cluster transport since an explanation of how 5 clusters moved under a shift is less complicated than an explanation of how 10 clusters moved. The above method allows us to have a partial ordering on interpretability without having to determine the absolute value of interpretability of an individual explanation $T$, as this requires expensive context-specific human evaluations, which is out of scope for this paper.

## 4    PRACTICAL METHODS FOR FINDING AND VALIDATING SHIFT EXPLANATIONS

In this section, we discuss practical methods for shift explanations. We first discuss using our $k$-sparse and $k$-cluster maps to allow a user to automatically change the level of interpretability of a shift explanation as desired. Coupled with a PercentExplained metric, this gives an operator various levels/complexities of explanation and a way to validate them. Next, we propose a practical approximation to Eqn. 1, the Interpretable Transport equation, and in Sections 4.3 and 4.4 we cover how to find the optimal explanation from $\Omega_{sparse}^{(k)}$ and $\Omega_{cluster}^{(k)}$ for this equation.

### 4.1    INTERPRETABILITY AS A HYPERPARAMETER

By optimizing Eqn. 1 we can find the best shift explanation for a given set of interpretable transport maps $\Omega$. However, directly defining a $\Omega$ which contains candidate mappings that are guaranteed to be both interpretable and expressive enough to explain a shift can be a difficult task. Thus, we can instead set $\Omega$ to be a super-class, such as $\Omega_{vector}$ given in subsection 3.2, and then adjust $k$ until a $\Omega^{(k)}$ is found which matches the needs of the situation. This allows a human operator to

request a mapping with better alignment by increasing $k$, which correspondingly will decrease the mapping's interpretability, or request a more interpretable mapping by decreasing the complexity (i.e. decreasing $k$) which will decrease the alignment.

To assist an operator in determining if the interpretability hyperparameter should be adjusted, we introduce a *PercentExplained* metric, which we define to be:

$$\text{PercentExplained}(P_{src}, P_{tgt}, T) := \frac{W_2^2(P_{src}, P_{tgt}) - W_2^2(T_\sharp P_{src}, P_{tgt})}{W_2^2(P_{src}, P_{tgt})} \tag{2}$$

where $W_2^2(\cdot, \cdot)$ is the squared Wasserstein-2 distance between two distributions. By rearranging terms (and ignoring the percentage scaling factor) we get $1 - \frac{W_2^2(T_\sharp P_{src}, P_{tgt})}{W_2^2(P_{src}, P_{tgt})}$, which shows this metric's correspondence to the statistics coefficient of determination $R^2$, where $W_2^2(T_\sharp P_{src}, P_{tgt})$ is analogous to the residual sum of squares and $W_2^2(P_{src}, P_{tgt})$ is similar to the total sum of squares. This gives an approximation of how much a current shift explanation $T$ accurately maps onto a target distribution. This can be seen as a normalization of a mapping's fidelity with the extremes being $T_\sharp P_{src} = P_{tgt}$, which fully captures a shift, and $T = \text{Id}$, which does not move the points at all. When provided this metric along with a shift explanation, an operator can decide whether to accept the explanation (e.g., the PercentExplained is sufficient and $T$ is still interpretable) or reject the explanation and adjust $k$.

### 4.2 EMPIRICAL INTERPRETABLE TRANSPORT

Since the divergence term in Eqn. 1 can be computationally-expensive to optimize in practice, we suggest an empirical approximation to the interpretable transport solution:

$$\underset{T \in \Omega}{\arg\min} \frac{1}{N} \sum_{i=1}^{N} c\left(\boldsymbol{x}^{(i)}, T(\boldsymbol{x}^{(i)})\right) + \lambda d\left(T(\boldsymbol{x}^{(i)}), T_{OT}(\boldsymbol{x}^{(i)})\right) \tag{3}$$

where $d$ is a distance function such as the $l_2$ distance or squared euclidean distance. Most notably, the divergence value in Eqn. 1 is replaced with the sum over distances between $T(\boldsymbol{x})$ and the optimal transport mapping for $\boldsymbol{x}$. This is computationally attractive as the optimal transport solution only needs to be calculated once, rather than calculating the Wasserstein distance once per iteration like in the Interpretable Transport solution (which even if the $W$-distance is approximated, can be expensive over many iterations). For optimization purposes, this is also reasonable since $\frac{1}{N} \sum_{i=1}^{N} d(T(\boldsymbol{x}^{(i)}), T_{OT}(\boldsymbol{x}^{(i)}))$ upper-bounds $\phi(P_{T(\boldsymbol{x})}, P_{tgt})$, when $d = \ell_2^2$, $\phi = W_2^2$ and $N$ approaches the population size of $P_{src}$ (proof shown in appendix).

### 4.3 FINDING $k$-SPARSE MAPS

Let $k$ be a desired level of interpretability; our goal is to find the optimal $k$ features to include in our active feature set $\mathcal{A}$ and then find the best transport along those features for a given transport class $\Omega$. A simple (and often ideal) approach to the feature selection problem is to select the $k$ features which have the largest average shift from the source distribution to the target distribution; this approach is used throughout this paper. Although the chosen $T$ will depend on the optimization over $\Omega$, we provide two closed-form solutions that give optimal alignment for a given $k$ under cases where $\Omega = \Omega_{vector}$ and when $\Omega$ is all possible mappings. The mapping which gives the best alignment in $\Omega_{vector}^{(k)}$ is $k$-sparse mean shift, i.e. $T(\boldsymbol{x}) = \boldsymbol{x} + \tilde{\mu}$ where $\tilde{\mu}$ is a vector where the $j^{\text{th}}$ coordinate is $[\mu_{tgt} - \mu_{src}]_j$, if $j \in \mathcal{A}$, else, it is 0. When $\Omega^{(k)}$ is all $k$-sparse functions, the shift explanation which minimizes the distance term in Eqn. 3 is the $k$-sparse optimal transport solution which sets each feature in $\mathcal{A}$ to match that of the OT push forward for that feature, i.e. $[T(\boldsymbol{x})]_j = [T_{OT}(\boldsymbol{x})]_j$ if $j \in \mathcal{A}$, else $[\boldsymbol{x}]_j$, thus allowing for arbitrary conditional transports for features in $\mathcal{A}$ (see proof **??**). The proofs for the two previous claims can be seen in the Appendix.

### 4.4 FINDING $k$-CLUSTER MAPS

Instead of shifting respective to features, we can define $k$ vector shifts for $k$ groups in our source domain, with the goal of explaining how each group changed from source to target. To do this, we perform *paired* clustering in the source and target domains, so that we can relate a given cluster in $P_{src}$ to its most similar counterpart in $P_{tgt}$ (as opposed to pushing the $k$ clusters in $P_{src}$ onto the entire target domain). With this, we construct $M_{src}$ and $M_{tgt}$ where the $k$ columns of $M$ represent the $k$ cluster means for the source and target distributions, respectively. Then, we define

$\Delta = M_{tgt} - M_{src}$ so that each vector shift $\delta_j$ is the difference in means between the $j^{\text{th}}$ source and the target clusters. In practice, the set of paired clusters can be found by performing clustering in a joint $Z$ space of $P_{src}$ and $P_{T_{OT}(\boldsymbol{x})}$ where the resultant $k$ cluster centroids in this space are of the form $[M_{src}, M_{tgt}]$. Formally, this is done using the algorithm seen in Alg. 1.

## 5 EXPERIMENTS

While in Appendix C we have experiments on simulated and simpler shifts which can be used to gain intuition on how the different shift explanation techniques work, in this section, we study the performance of our methods when applied to real-world data. We first present results using $k$-sparse transport as our method of explaining shifts between toxic and non-toxic comments across splits from the Stanford WILDS distribution shift benchmark Koh et al. (2021) version of the "CivilComments" Dataset Borkan et al. (2019). We then, use $k$-cluster transport to explain the difference between different groups of the male population and groups of the female population in the U.S. Census "Adult Income" dataset Kohavi & Becker (1996). Finally, in the next section, we provide a general framework for explaining image-based shifts in high dimensional regimes (e.g., images).

Table 1: A baseline vanilla mean shift explanation, $k$-sparse mean shift explanation, ($k$-$\mu$-Ex), and $k$-sparse OT explanations ($k$-OT-Ex) for the three splits from CivilComments (to save space the baseline is only used for F$\rightarrow$M). Each cell represents adding/subtracting a unigram from $P_{src}$ to align it with the comment distribution of $P_{tgt}$ and the respective PercentExplained (excluding the baseline method). For example, in $k$-$\mu$-Ex(F$_0\rightarrow$F$_1$), adding "stupid" aligns the non-toxic female comments to the toxic female comments and cumulatively explains $0.2\%$ of the shift.

| $k$ | Baseline: $\mu(F, M)$ | $k$ | $k$-$\mu$-Ex(F, M) | | $k$-OT-Ex(F, M) | | $k$ | $k$-$\mu$-Ex(F$_0$, F$_1$) | | $k$-OT-Ex(F$_0$, F$_1$) | | $k$ | $k$-$\mu$-Ex(M$_0$, M$_1$) | | $k$-OT-Ex(M$_0$, M$_1$) | |
|---|---|---|---|---|---|---|---|---|---|---|---|---|---|---|---|---|
| 1 | + man | 1 | + man | 2.4% | + man | 6.9% | 1 | + white | 0.1% | + trump | 2.6% | 1 | + trump | 0.2% | - women | 2.8% |
| 2 | + men | 2 | + men | 2.4% | + men | 9.2% | 2 | - women | 0.1% | + people | 3.7% | 2 | - women | 0.3% | - men | 4.1% |
| 3 | - woman | 3 | - woman | 2.6% | - woman | 10.8% | 3 | + like | 0.2% | + woman | 4.7% | 3 | + black | 0.3% | + man | 5.3% |
| 4 | + white | 4 | + white | 2.7% | + white | 12.0% | 4 | + stupid | 0.2% | + like | 5.7% | 4 | - church | 0.3% | + people | 6.3% |
| ... [29K more entries] ... | | 5 | + male | 2.8% | - people | 13.0% | 5 | - church | 0.2% | - men | 6.6% | 5 | + stupid | 0.3% | + like | 7.3% |
| 29,553 | + martina | 6 | + black | 2.8% | - like | 13.9% | 6 | + hillary | 0.2% | + just | 7.5% | 6 | + gay | 0.4% | + trump | 8.3% |
| 29,554 | - diqlmjawsae | 7 | + god | 2.8% | - just | 14.7% | 7 | + black | 0.2% | + don't | 8.3% | 7 | + racist | 0.4% | + just | 9.2% |
| 29,555 | - да | 8 | - female | 2.8% | + male | 15.4% | 8 | + sex | 0.2% | + white | 9.2% | 8 | - god | 0.4% | + don't | 10.9% |
| 29,556 | - bodybuilder | 9 | - abortion | 2.8% | - don't | 16.3% | 9 | - female | 0.2% | + man | 9.9% | 9 | - jesus | 0.4% | + black | 11.6% |
| 29,557 | + philhiblers | 10 | + males | 2.8% | + god | 16.8% | 10 | - abortion | 0.3% | - think | 10.5% | 10 | + man | 0.4% | - male | 12.3% |

**Civil Comments Dataset** Here we present results using $k$-sparse shifts to explain the difference between three splits of the CivilComments dataset Borkan et al. (2019) from the WILDS datasets Koh et al. (2021). This dataset consists of comments scraped from the internet where each comment is paired with a binary toxicity label and demographic information pertaining to the content of the comment. If we were an operator trying to see how the comments and their toxicity change across targeted demographics, we could create three splits: {F, M}, {F$_0$, F$_1$}, and {M$_0$, M$_1$}, where F represents all female comments, M are all male comments, and F$_0$, F$_1$ are nontoxic, toxic female comments, respectively (and likewise for males). We can explain these three splits using vanilla mean shift, a $k$-sparse mean shift ($k$-$\mu$), and $k$-sparse OT ($k$-OT) shift explanations, as seen in Table 1 which shows results for the unigrams which the maximize the alignment between the unigram distributions created for each split. The baseline vanilla mean-shift explanation yields all 30K features at once (with no guide for truncating), while the $k$-sparse shifts provide explanations up to a $k$ words as well as a corresponding PercentExplained to aid in determining if additional words should be added to the explanation. Note that for $k$-$\mu$ explanations, when transporting a word, that word is added equally to all comments in $P_{src}$, while since $k$-OT allows for each comment to be shifted optimally (via conditioning on the other words in each comment), thus $k$-OT can explain significantly more of the shift by transporting the same word.

It is clear that performing a vanilla full mean shift explanation on the unigram data between splits is unwise due to the high dimensionality of the data and, without further information, it is unclear when to truncate such an explanation. However, in our approach, by iteratively reporting the shifted unigrams along with the cumulative PercentExplained, a practitioner can better understand the impact each additional word has on the shift explanation. For example, it makes sense that adding "man", "men", and subtracting "woman" were the three unigrams that best aligned the female and male comment distributions and could account for as much as 10% of the shift. By moving from nontoxic to toxic comments we can see a decrease in plural words such as "women" and "men" suggesting individuals are being targeted, and when narrowing to male nontoxic to toxic mappings we see an increase in words such as "gay" and "racist" as opposed to F$_{0\rightarrow1}$ which shows an increase in words such as "sex" and "white".

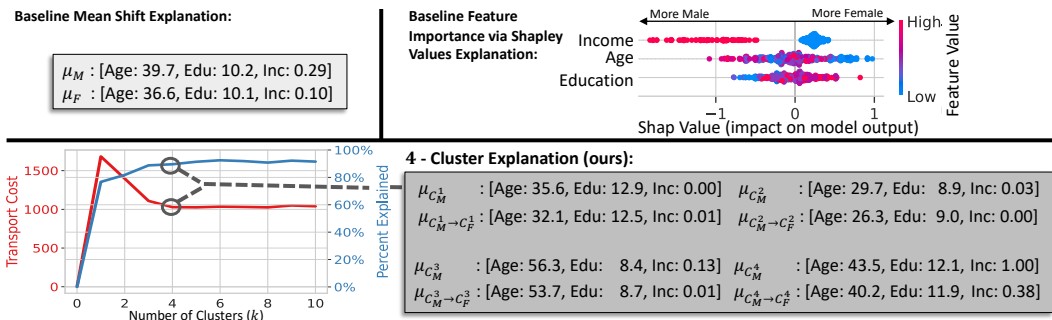

Figure 2: Using $k$-cluster transport (bottom) to explain the shift from the male population to the female population of the Adult Income dataset allows us to capture how heterogeneous groups within the dataset moved. For example, our cluster-based explanation (where $k = 4$) highlights the larger income disparity between males and females with a bachelor's degree (edu=12) seen in $C^4$, which is hidden in the mean shift explanation. On the other hand, comparing the means of the two distributions (top left) or the Shapley values calculated from a Random Forest classifier trained to classify between the Male/Female distributions (top right) only allows us to see that the income gap as a predominate difference, but provide no additional actionable detail.

**Adult Income Dataset**    This dataset originally comes from the United States 1994 Census database and is commonly used to predict whether a person's annual income exceeds $50k using 14 demographic features. Similar to Budhathoki et al. (2021), we consider a subset of non-redundant features: age, years of education (where 12+ is beyond high school), and income (which is encoded as 1 if the person's annual income is greater than $50k and 0 if it is below). We then split this dataset along the sex dimension, and define our source distribution as the male population and the target as the female population. In order to find the set of paired clusters, we first standardize a copy of the data to have zero mean and unit variance across all features, where the $\mu$ and $\sigma$ used for the standardization are found via the feature-wise mean and standard deviation of the source distribution and perform clustering in the standardized joint space using the method described in Section 4.4. The $k$ clustering labels are then used to label points to clusters in the original (unstandardized) data space.

Let's assume the role of a researcher looking to implement a social program targeting gender inequalities. We could compare the means of the male/female distributions which shows on average a 20% lower chance of having an annual income above $50k when moving from the male population to the female population. Additionally, we could train a classifier to predict between male/female data points and use a feature importance measurement tool like Shapley values Lundberg & Lee (2017) to determine that income is a main differing feature. However, let's say we want to dig deeper. We could instead use $k$-cluster transport to see how heterogeneous subgroups shifted across a range of clusters, as seen in Fig. 2. If we accept the explanation at $k = 4$ (since beyond this, the marginal advantage of adding an additional cluster is minimal in terms of both transport cost and PercentExplained), with this more detailed explanation we could see in $\mu_{C_M^4}$ the more significant drop in high-income likelihood between middle-age adults with a bachelors degree (from males having nearly a 100% likelihood of having an income $\geq$50k to only a 38% chance when pushed onto the female cluster). This sub-shift, which was hidden in both the mean-shift and distribution classifier explanations, gives us a significantly narrower scope as a starting point to look deeper at the discrepancies between a given population of men and women.

## 6    EXTENDING TO EXPLAINING SHIFTS IN IMAGES

Thus far, our focus has been on building a foundation for explaining distribution shifts via intrinsically interpretable transportation maps and showing the baseline efficacy of this approach. As seen in the Stanford Wilds dataset Koh et al. (2021), which contains benchmark examples of real-world image-based distribution shifts, image-based shifts can be immensely complex and context-specific–even when the oracle shift is known. In order to provide an adequate intrinsically interpretable mapping explanation of a distribution shift in image data, multiple new advancements must first be met (e.g., finding a disentangled latent space with semantically meaningful dimensions, approximating high dimensional empirical optimal transport maps, etc.), which are out of scope of this paper. Thus, we first present an initial framework for finding intrinsically interpretable transport maps to explain

distribution shifts in images, which we hope future work can build upon. Then, in subsection 6.2, we use visualizations of complex distributional mappings as a post-hoc transport map interpretability method which alleviates the intrinsic interpretability requirement (allowing for more expressive mappings), but leaves more up to operator interpretation.

## 6.1 HIGH DIMENSIONAL INTERPRETABLE TRANSPORT MAPS

In order to find interpretable transport mappings for high dimensional spaces like images, we can project $P_{src}$ and $P_{tgt}$ onto an *interpretable* latent space (e.g., a space which has disentangled and semantically meaningful dimensions) which is learned by some (pseudo-)invertible function $g : \mathbb{R}^d \rightarrow \mathbb{R}^k$ where $k < d$ (e.g., an autoencoder). Then, we can solve for an interpretable mapping such that it aligns the distributions in the latent space, $P_{T(g(\boldsymbol{x}))} \approx P_{g(\boldsymbol{y})}$. For counterfactual purposes, we can use $g^{-1}$ to project $T(g(\boldsymbol{x}))$ back to $\mathbb{R}^d$ to display the transported image to an operator. With this, we can define our set of high dimensional interpretable transport maps: $\Omega_{\text{high-dim}} := \left\{ T : T = g^{-1}\left(\tilde{T}(g(\boldsymbol{x}))\right), \tilde{T} \in \Omega, g \in \mathcal{I} \right\}$ where $\Omega$ the set of interpretable mappings (e.g., $k$-sparse mappings) and $\mathcal{I}$ is the set of (pseudo-)invertible functions with an interpretable (i.e. semantically meaningful) latent space. Finally, given an interpretable $g \in \mathcal{I}$, this gives us *High-dimensional Interpretable Transport*, $T_{HIT}$:

$$\arg\min_{\tilde{T} \in \Omega^{(k)}} \mathbb{E}_{P_{src}}\left[ c\left(g(\boldsymbol{x}), \tilde{T}(g(\boldsymbol{x}))\right)\right] + \lambda\phi(P_{\tilde{T}(g(\boldsymbol{x}))}, P_{g(\boldsymbol{y})}) \tag{4}$$

which results in an interpretable map $\tilde{T}$ which approximately shows how images from $P_{src}$ shifted to $P_{tgt}$ in a semantically meaningful way (e.g., how the H&E staining in histopathology images changes across hospitals). For further details about $T_{HIT}$, its variants, and results for an experiment on explaining Colorized-MNIST, please see Appendix D.

## 6.2 EXPLAINING IMAGE-BASED SHIFTS VIA COUNTERFACTUAL EXAMPLES

In some cases, solving for an interpretable latent space can be too difficult/costly, and thus a shift cannot be expressed by an interpretable mapping function. However, if the samples themselves are easy to interpret (e.g., images), we can still explain a transport mapping by visualizing translated samples. Specifically, we can remove the interpretable constraint on the mapping and leverage methods from the unpaired Image-to-Image translation (I2I) literature to translate between the source and target domain while preserving the content. For a comprehensive summary of the recent I2I works and methods, please see Pang et al. (2021). Once a distributional mapping is found, to serve as an explanation, we can provide an operator with a set of counterfactual pairs $\{(\boldsymbol{x}, T(\boldsymbol{x})) : \boldsymbol{x} \sim P_{src}, T(\boldsymbol{x}) \sim P_{tgt}\}$. Then, by determining what commonly stays invariant and what commonly changes across the set of counterfactual pairs, this can serve as an explanation of how the source distribution shifted to the target distribution. While more broadly applicable, this approach could put a higher load on the operator than an intrinsically interpretable mapping approach.

**Explaining Shifts in H&E Images Across Hospitals** We apply this distribution counterfactual approach to the Camelyon17 dataset Bandi et al. (2018) which is a real-world distribution shift dataset that consists of whole-slide histopathology images from five different hospitals. We use the Stanford WILDS Koh et al. (2021) variant of the dataset which converts the whole-slide images into over 400 thousand patches. Since each hospital has varying hematoxylin and eosin (H&E) staining characteristics, this, among other batch effects, leads to heterogeneous image distributions across hospitals as can be seen in Fig. 3.

To generate the counterfactual examples, we treat each hospital as a domain and train a StarGAN model Choi et al. (2018) to translate between each domain. For training, we followed the original training approach seen in Choi et al. (2018), with the exception that we perform no center cropping. After training, we can generate distribution counterfactual examples by inputting a source image and the label of the target hospital domain to the model. Counterfactual generation was done for all five hospitals and can be seen on the right-hand side of Fig. 3. It can be seen that the StarGAN model captures the different staining characteristics across the hospitals. For example, hospital 1 ($P_1$) consists of mostly light staining, and thus transporting to this domain usually involves lightening of the image while $P_3$ seems to have more hematoxylin stain thus leading to deeper purple images when pushing onto this domain. We can also see that the model tends to respect the content of the image where patches that contain tumor cells (e.g., the $P_5$ sample on the right-hand side) still

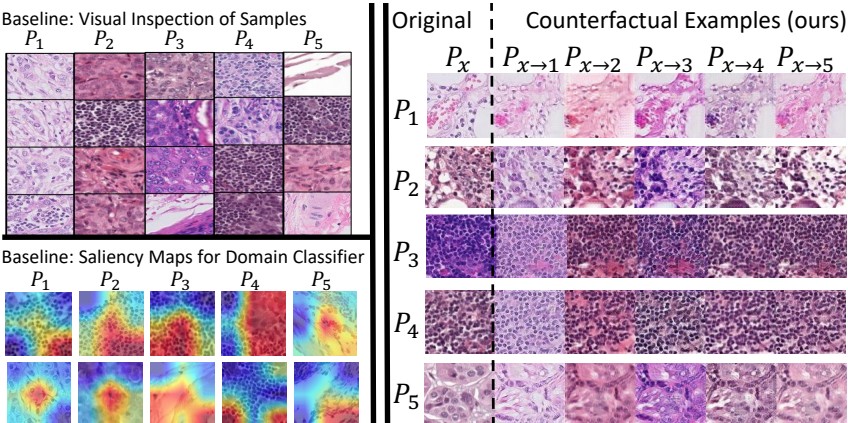

Figure 3: Our explanation approach (right) of showing paired counterfactual images translated between the hospital domains (represented as $P_1$, $P_2$, ...) quickly makes it clear how the staining/coloring differs across the hospital domains (where the $(i, j)$ row, column pair represents the pushforward of the i-th domain onto the j-th domain). The baseline method of unpaired samples (top-left) requires many more samples to begin to understand the differences across the hospital domains and using Grad-CAM Selvaraju et al. (2016) to explain a ResNet-50 He et al. (2016) domain classifier (bottom-left) does not capture the appropriate staining/coloring details.

contain tumor cells in the counterfactual cases and likewise for lymphocyte cells (e.g., the $P_4$ sample on the right-hand side). We further explore content-based changes (as opposed to style changes) of an image using the CelebA dataset in subsection C.4.

## 7 LIMITATIONS

A primary challenge in developing distribution shift explanations is determining how to evaluate the efficacy of a given explanation in a given context. Evaluating explanations is an active area of research Robnik-vSikonja & Bohanec (2018); Molnar (2020); Doshi-Velez & Kim (2017) with commonalities such as an explanation should be contrastive, succinct, should highlight abnormalities, and should have high fidelity Molnar (2020). For the case of distribution shift explanations, as this is a highly context-dependent problem (dependent on the data setting, task setting, and operator knowledge) and our approach is designed to tackle this problem in general, we do not have a general automated way of measuring whether a given explanation is indeed interpretable. Instead, we provide a general contrastive method that supplies the PercentExplained (approximation of fidelity) and the adjustable $k$-level of sparse/cluster mappings (which trades off between succinctness and fidelity) but ultimately leaves the task of validating the explanation up to the operator. We believe developing new shift explanation maps and criteria for specific applications (e.g., explaining the results of experiments run with different initial conditions), thus offering tighter interpretability measurement bounds, is a rich area for future work. An additional challenge is that while the PercentExplained metric shows the fidelity of an explanation (i.e. how aligned $T_\sharp(P_{src})$ and $P_{tgt}$ are), we do not have a method of knowing specifically what is missing from the explanation – which can be considered a "known unknown". For further discussions of challenges with explaining distribution shifts (e.g., finding an interpretable latent space, approximations of Wasserstein distances in high dimensional regimes, etc.) we point the reader to Appendix B.

## 8 DISCUSSION AND CONCLUSION

In this paper, we introduced a framework for explaining distribution shifts using a transport map $T$ between a source and target distribution. We constrained a relaxed form of optimal transport to theoretically define an interpretable mapping $T_{IT}$ and introduced two interpretable transport methods: $k$-sparse and $k$-cluster transport. We provided practical approaches to calculating a shift explanation, which allows us to use treat interpretability as a hyperparameter that can be adjusted based on a user's need and showed how our methods can help an operator investigate a distribution shift on real-world examples.

We hope our work suggests multiple natural extensions such as using trees as a feature-axis-aligned form of clustering or even other forms of interpretable sets. Given our results and potential ways forward, we ultimately hope our framework lays the groundwork for providing more information to aid in investigations of distribution shift.

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

# A  ADDITIONAL TRANSPORTATION MAPPINGS DETAILS AND PROOFS

## A.1  FINDING K-PAIRED CLUSTERS

---
**Algorithm 1** Finding $k$ Paired Clusters

---
**Input:** $X, Y, k$
$d \leftarrow X.ndim$
$T_{OT} \leftarrow \text{OptimalTransportAlg}(X, Y)$ //e.g., Sinkhorn
$Z \leftarrow [X, T_{OT}(X)]$
$Z_{cluster-centroids} \leftarrow \text{ClusteringAlg}(Z, k)$ //e.g., k-means
$M_{src} \leftarrow [Z_{cluster-centroids}]_{1:d}$ //slicing column-wise
$M_{tgt} \leftarrow [Z_{cluster-centroids}]_{d:2d}$
**Output:** $M_{src}, M_{tgt}$

---

## A.2  INFINITE NUMBER OF POSSIBLE MAPPINGS BETWEEN DISTRIBUTIONS

As stated in the introduction, given two distributions, there exist many possible mappings such that $T_\sharp P_{src} = P_{tgt}$ (it should be noted that here we are speaking of the general mapping problem, not the *optimal* transport problem which can be shown via Brenier's theorem Peyré & Cuturi (2019) to have a unique matching for some cases). For instance, given two isometric Gaussian distributions $\boldsymbol{x} \sim \mathcal{N}_1(\mu_1, I)$, $\boldsymbol{y} \sim \mathcal{N}_2(\mu_2, I)$, where $I$ is the Identity matrix, there exist an infinite number of $T$'s such that $T(\boldsymbol{x}) \sim \mathcal{N}_2$. Specifically, any $T$ of the form: $T(\boldsymbol{x}) = \mu_2 + R(\boldsymbol{x} - \mu_1)$, where is $R$ is an arbitrary rotation matrix, will shift $T_\sharp \mathcal{N}_1$ to have a mean of $\mu_2$ and perfectly align the two distributions (since any rotation of an isometric Gaussian will still be an isometric Gaussian).

## A.3  DISTANCE IN EMPIRICAL INTERPRETABLE TRANSPORT UPPER-BOUNDS THE WASSERSTEIN DISTANCE

First, let's remember our empirical method for finding $T$:

$$\underset{T \in \Omega}{\arg\min} \; \frac{1}{N} \sum_i^N c(\boldsymbol{x}^{(i)}, T(\boldsymbol{x}^{(i)})) + \lambda d(T(\boldsymbol{x}^{(i)}), T_{OT}(\boldsymbol{x}^{(i)})) \tag{5}$$

where $T_{OT}$ is the optimal transport solution between our source and target domains with the given $c$ cost function. The distance term $d$ on the right-hand side of this equation is assumed to be the $\ell_2$ cost or squared euclidean cost and is an empirical approximation of the divergence term $\phi(P_{T(\boldsymbol{x})}, P_Y)$ in Eqn. 1, where $\phi$ is assumed to be the Wasserstein distance, $W$. We claim this is a reasonable approximation since as $N$ approaches the size of the dataset (or for densities, $\lim_{N \to \infty}$), the distance term becomes the expectation: $\mathbb{E}_{x \sim P_{src}} d(T(\boldsymbol{x}^{(i)}), T_{OT}(\boldsymbol{x}^{(i)}))$ which is an upper-bound on the $W(P_{T(\boldsymbol{x})}, P_Y)$ distance. To show this, we start with the expanded $W$ distance:

$$W(P_{T(\boldsymbol{x})}, P_Y) = \min_{R \in \Psi} \mathbb{E}_{\boldsymbol{x} \sim P_{src}} d\left(T(\boldsymbol{x}), R(T(\boldsymbol{x}))\right), \quad \Psi := \{R : R_\sharp T(\boldsymbol{x}) = P_Y\}$$

$$\leq \mathbb{E}_{\boldsymbol{x} \sim P_{src}} d\left(T(\boldsymbol{x}), R(T(\boldsymbol{x}))\right), \qquad \forall R \in \Psi$$

$$\text{If we let } Q = T_{OT} \cdot T^{-1}, \text{ and since } Q \in \Psi \text{ we can say}$$

$$\leq \mathbb{E}_{\boldsymbol{x} \sim P_{src}} d\left(T(\boldsymbol{x}), Q(T(\boldsymbol{x}))\right) = \mathbb{E}_{\boldsymbol{x} \sim P_{src}} d\left(T(\boldsymbol{x}), T_{OT}(\boldsymbol{x})\right)$$

$$\implies W(P_{T(\boldsymbol{x})}, P_Y) \leq \mathbb{E}_{\boldsymbol{x} \sim P_{src}} d\left(T(\boldsymbol{x}), T_{OT}(\boldsymbol{x})\right)$$

## A.4  PROVING THE K-SPARSE OPTIMAL TRANSPORT IS THE K-SPARSE TRANSPORT THAT MINIMIZES OUR DISTANCE FROM OT LOSS

When performing unrestricted $k$-sparse transport, i.e. where $\Omega_{sparse}^{(k)}$ is any transport which only moves points along $k$ dimensions, the $k$-sparse optimal transport solution is the exact mapping that minimizes the distance function in the right-hand side of Eqn. 5 if $d$ is the $\ell_2$ distance or squared Euclidean distance. As a reminder, $k$-sparse optimal transport is: $[T(\boldsymbol{x})]_j = [T_{OT}(\boldsymbol{x})]_j$ if $j \in \mathcal{A}$,

else $[\boldsymbol{x}]_j$, where $\mathcal{A}$ is the active set of $k$ dimensions which our $k$-sparse transport $T$ can move points. Let $\bar{\mathcal{A}}$ be $\mathcal{A}$'s complement (i.e. the dimensions which are unchanged under $T$). Let $\boldsymbol{z} = T(\boldsymbol{x})$, $\boldsymbol{z}_{OT} = T_{OT}(\boldsymbol{x})$, and $\boldsymbol{x} \in \mathbb{R}^{n \times d}$. If $d$ is the squared Euclidean distance:

$$
\begin{aligned}
d(\boldsymbol{z}, \boldsymbol{z}_{OT}) &= \sum_{j \in [d]} \sum_{i \in [n]} \left( \boldsymbol{z}_{i,j} - \boldsymbol{z}_{OT_{i,j}} \right)^2 \\
&= \sum_{j \in \mathcal{A}} \sum_{i \in [n]} \left( \boldsymbol{z}_{i,j} - \boldsymbol{z}_{OT_{i,j}} \right)^2 + \underbrace{\sum_{j \in \bar{\mathcal{A}}} \sum_{i \in [n]} \left( \boldsymbol{x}_{i,j} - \boldsymbol{z}_{OT_{i,j}} \right)^2}_{= \alpha \text{ , since constant w.r.t T}} \\
&= \sum_{j \in \mathcal{A}} \sum_{i \in [n]} \left( \boldsymbol{z}_{i,j} - \boldsymbol{z}_{OT_{i,j}} \right)^2 + \alpha \\
&\quad \text{now if T is the truncated optimal transport solution, } [\boldsymbol{z}]_j = [\boldsymbol{z}_{OT}]_j \quad \forall j \in \mathcal{A} \\
&= 0 + \alpha
\end{aligned}
$$

Since $\alpha$ is the minimum of $d(\boldsymbol{z} - \boldsymbol{z}_{OT})$ for a given $\mathcal{A}$, the truncated optimal transport problem minimizes the $d(T(\boldsymbol{x}^{(i)}), T_{OT}(\boldsymbol{x}^{(i)}))$ distance. This can easily be extended to show that the optimal active set for this case is the one that minimizes $\alpha$, thus the active set should be the $k$ dimensions which have the largest squared difference between $\boldsymbol{x}$ and $\boldsymbol{z}_{OT}$.

### A.5 PROOF THAT K-MEAN SHIFT IS THE K-VECTOR SHIFT THAT GIVES US THE BEST ALIGNMENT

When performing $k$-sparse vector transport, i.e. where $\Omega^{(k)}_{vector} = \{T : T(\boldsymbol{x}) = \boldsymbol{x} + \tilde{\delta}\}$ where $\tilde{\delta} = [\delta]_j$ if $j \in \mathcal{A}$ else $[\delta]_j = 0$ and $\delta \in \mathbb{R}^d$, $|\mathcal{A}| \leq k$, the $k$-sparse mean shift solution is the exact mapping that minimizes the distance function in the right-hand side of Eqn. 5 when $d$ is the $\ell_2$ distance.

$$
\begin{aligned}
d(\boldsymbol{z}, \boldsymbol{z}_{OT}) &= \sum_{j \in [d]} \sum_{i \in [n]} \left( \boldsymbol{z}_{i,j} - \boldsymbol{z}_{OT_{i,j}} \right)^2 \\
&= \sum_{j \in \mathcal{A}} \sum_{i \in [n]} \left( \boldsymbol{z}_{i,j} - \boldsymbol{z}_{OT_{i,j}} \right)^2 + \underbrace{\sum_{j \in \bar{\mathcal{A}}} \sum_{i \in [n]} \left( \boldsymbol{x}_{i,j} - \boldsymbol{z}_{OT_{i,j}} \right)^2}_{= \alpha \text{ , since constant w.r.t T}} \\
&= \sum_{j \in \mathcal{A}} \sum_{i \in [n]} \left( \boldsymbol{z}_{i,j} - \boldsymbol{z}_{OT_{i,j}} \right)^2 + \alpha \\
&= \sum_{j \in \mathcal{A}} \sum_{i \in [n]} \left( \boldsymbol{x}_{i,j} + \delta_j - \boldsymbol{z}_{OT_{i,j}} \right)^2 + \alpha \\
&= \sum_{j \in \mathcal{A}} \sum_{i \in [n]} \left( \boldsymbol{x}_{i,j}^2 + \delta_j^2 + \boldsymbol{z}_{OT_{i,j}}^2 + 2\delta_j(\boldsymbol{x}_{i,j} - \boldsymbol{z}_{OT_{i,j}}) - 2\boldsymbol{z}_{OT_{i,j}}\delta_j - 2\boldsymbol{x}_{i,j}\boldsymbol{z}_{OT_{i,j}} \right) + \alpha
\end{aligned}
$$

Similar to the $k$-sparse optimal transport solution, we can see that $\mathcal{A}$ should be selected as the $k$ dimensions which have the largest shift, thus minimizing $\alpha$. The coordinate-wise gradient of the above equation is:

$$
\nabla_{\delta_j} d(\boldsymbol{z}, \boldsymbol{z}_{OT}) = \begin{cases} \sum_{i \in [n]} \left( 2\delta_j + 2\boldsymbol{x}_{i,j} - 2\boldsymbol{z}_{OT_{i,j}} \right) & j \in \mathcal{A} \\ 0 & j \in \bar{\mathcal{A}} \end{cases}
$$

Now with this we can say:

$$\nabla_{\delta_{j} \in \mathcal{A}} \, d(\boldsymbol{z}, \boldsymbol{z}_{OT}) = \sum_{i \in [n]} \left( 2\delta_j + 2\boldsymbol{x}_{i,j} - 2\boldsymbol{z}_{OT_{i,j}} \right)$$

$$= 2n\delta_j + \sum_{i \in [n]} \left( 2\boldsymbol{x}_{i,j} - 2\boldsymbol{z}_{OT_{i,j}} \right)$$

$$\text{now let } \delta_j = \delta_j^*$$

$$0 = 2n\delta_j^* + \sum_{i \in [n]} \left( 2\boldsymbol{x}_{i,j} - 2\boldsymbol{z}_{OT_{i,j}} \right)$$

$$n\delta_j^* = \sum_{i \in [n]} \left( \boldsymbol{z}_{OT_{i,j}} - \boldsymbol{x}_{i,j} \right)$$

$$\delta_j^* = \frac{1}{n} \sum_{i \in [n]} \left( \boldsymbol{z}_{OT_{i,j}} - \boldsymbol{x}_{i,j} \right)$$

$$\delta_j^* = \mu_{\boldsymbol{z}_{OT_j}} - \mu_{\boldsymbol{x}_j}$$

Thus showing the optimal delta vector to minimize $k$-vector transport is exactly the $k$-sparse mean shift solution.

## B   CHALLENGES OF EXPLAINING DISTRIBUTION SHIFTS AND LIMITATIONS OF OUR METHOD

Distribution shift is a ubiquitous and quite challenging problem. Thus, we believe discussing the challenges of this problem and the limitations of our solution should aid in advancements in this area of explaining distribution shifts.

As mentioned in the main body, as distribution shifts can take many forms, trying to explain a distribution shift is a highly context-dependent problem (i.e., dependent on the data setting, task setting, and operator knowledge). Thus, a primary challenge in developing distribution shift explanations is determining how to evaluate whether a given explanation is valid for a given context. In this work, we hope to introduce the problem of explaining distribution shifts *in general* (i.e. not with a specific task nor setting in mind), therefore we do not have an automated way of measuring whether a given explanation is indeed interpretable. Evaluating explanations is an active area of research Robnik-vSikonja & Bohanec (2018); Molnar (2020); Doshi-Velez & Kim (2017) with commonalities such as an explanation should be contrastive, succinct, should highlight abnormalities, and should have high fidelity. Instead, we introduce a proxy method that supplies the operator with the PercentExplained and the adjustable $k$-level of sparse/cluster mappings but leaves the task of validating the explanation up to the operator. We believe developing new shift explanation maps and criteria for specific applications (e.g., explaining the results of experiments run with different initial conditions) is a rich area for future work.

Explaining distribution shifts becomes more difficult when the original data is not interpretable. This typically can take two forms: 1) the raw data *features* are uninterpretable but the samples are interpretable (e.g., a sample from the CelebA dataset Liu et al. (2015) is interpretable but the pixel-level features are not) or 2) when both the raw data features and samples are uninterpretable (e.g., raw experimental outputs from material science simulations). In the first case, one can use the set of counterfactual pairs method outlined in subsection 6.2 (see Fig. 8 for examples with CelebA), however, as mentioned in the main paper, this is less sample efficient than an interpretable transport map. For the second case, if the original features are not interpretable, one must first find an interpretable latent feature space – which is a challenging problem by itself. As seen in Fig. 10, it is possible to solve for a semantic latent space and solve interpretable transport maps within the latent space, in this case, the latent space of a VAE model. However, if the meaningful latent features are not extracted, then any transport map within this latent space will be meaningless. In the case of Fig. 10, the 3-cluster explanation is likely only interpretable because we know the ground truth and thus know what to look for. As such, this is still an open problem and one we hope future work can improve on.

Additionally, while the PercentExplained metric shows the fidelity of an explanation (i.e. how aligned $T_\sharp(P_{src})$ and $P_{tgt}$ are), we do not have a method of knowing specifically *what* is missing from the explanation. This missing part of the explanation can be considered a "known unknown". For example, if a given $T$ has a PercentExplained of 85%, we know how much is missing, but we do not know what information is contained in the missing 15%. Similarly, when trying to explain an image-based distribution shift with large differences in content (e.g., a dataset with blonde humans and a dataset with bald humans), leveraging existing style transfer architectures (where one wishes to only change the style of an image while retaining as much of the original content as possible) to generate distributional counterfactuals can lead to misleading explanations. This is because explaining image-based distribution shifts might require large changes in content (such as removing head hair from an image), which most style-transfer models are biased against doing. As an example, Fig. 8 shows an experiment that translates between five CelebA domains (blond hair, brown hair, wearing hat, bangs, bald). It can be seen that the StarGAN model can successfully translate between stylistic differences such as "blond hair" → "brown hair" but is unable to make significant content changes such as "bangs" → "bald".

The above issues are mainly problems that affect distribution shift explanations in general, but below are issues specific to our shift explanation method (or any method which similarly uses empirical OT). Since we rely on the empirical OT solution for the sparse and cluster transport (and the percent explained metric), the weaknesses of empirical OT are also inherited. For example, empirical OT, even using the Sinkhorn algorithm with entropic regularization, scales at least quadratically in the number of samples Cuturi (2013). Thus, this is only practical for thousands of samples. Furthermore, empirical OT is known to poorly approximate the true population-level OT in high dimensions although entropic regularization can reduce this problem Genevay et al. (2019). Finally, empirical OT does not provide maps for new test points. Some of these problems could be alleviated by using recent Wasserstein-2 approximations to optimal maps via gradients of input-convex neural networks based on the dual form of Wasserstein-2 distance Korotin et al. (2019); Makkuva et al. (2020). Additionally, when using $k$-cluster maps, the clusters are not guaranteed to be significant (i.e. it might be indiscernible what makes this cluster different than another cluster), and thus if using $k$-cluster maps on datasets that do not have natural significant clusters (e.g., $P_{src} \sim$ Uniform$(0, 1)$, $P_{tgt} \sim$ Uniform$(1, 2)$) an operator might waste time looking for significance where there is none. While this cannot be avoided in general, using a clustering method that is either specifically designed for finding interpretable clusters Fraiman et al. (2013); Bertsimas et al. (2021) or one which directly optimizes the objective in interpretable transport equation Eqn. 1 might lead to easier to explanations which are easier to interpret or validate.

## C EXPERIMENTS ON KNOWN SHIFTS

Here we present additional results on simulated experiments as well as an experiment on UCI "Breast Cancer Wisconsin (Original)" dataset Mangasarian & Wolberg (1990). Our goal is to illuminate when to use the different sets of interpretable transport, and how the explanations can be interpreted, where in this case, a ground truth explanation is known. [1]

### C.1 SIMULATED EXPERIMENTS

In this section we study three toy shift problems: a mean shift between two, otherwise identical, Gaussian distributions, a Gaussian mixture model where each mixture component has a different mean shift, and a flipped and shifted half-moon dataset, as seen in figures (a), (d), and (g) in Fig. 4.

The first case is of a mean shift between two, otherwise identical, Gaussian distributions can be easily explained using $k$-sparse mean shift (as well as vanilla mean shift). We first calculate the OT mapping $T_{OT}$ between the two Gaussian distributions, which has a closed form solution of $T_{OT}(\boldsymbol{x}) = \mu_{tgt} + A(\boldsymbol{x} - \mu_{src})$, where $A$ is a matrix that can be seen as a conversion between the source and target covariance matrices, and because the covariance matrices are identical, A is the identity.

---

[1] Code to recreate all experiments seen here and in the main body of the paper will be released upon acceptance.

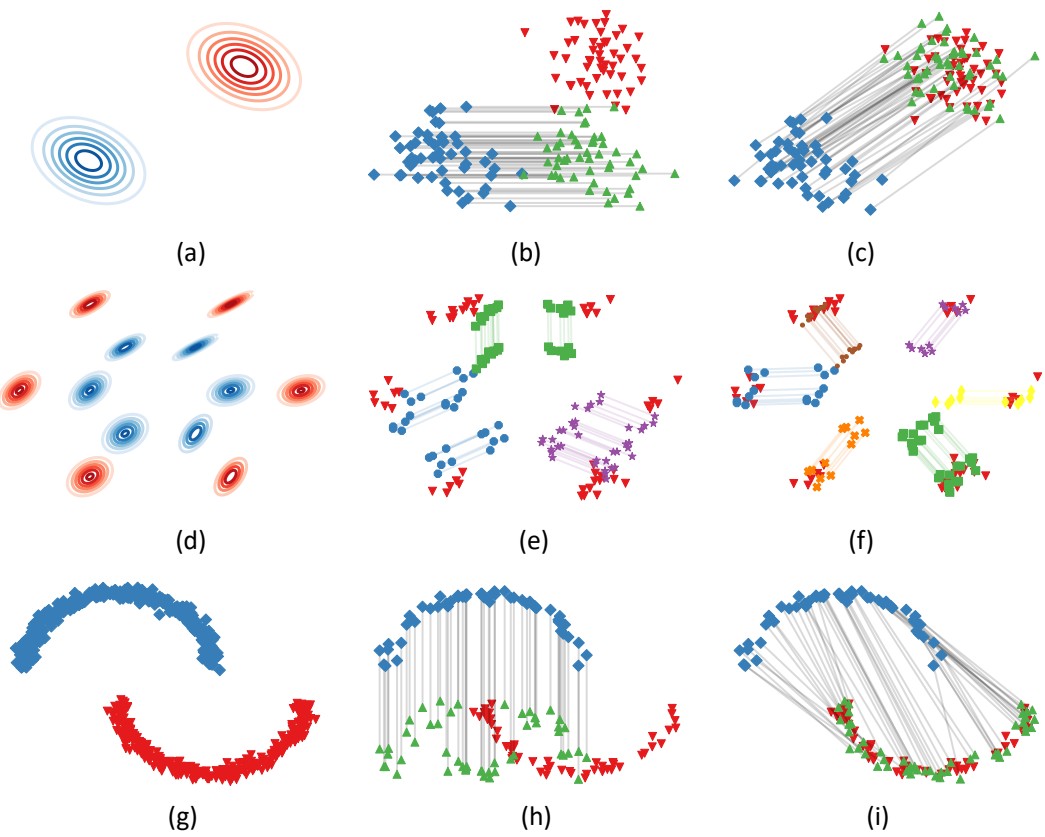

Figure 4: Three toy dataset shift examples showing the advantages of the different shift explanation methods, where a mean shift between Gaussians (top row) can be easily explained using $k$-sparse vector shifts, a varying mean shift across mixture components of a Gaussian mixture model (middle row) is best explained using $k$-sparse transport maps, while a complex shift (bottom row) requires a complex feature-wise mapping, such as $k$-sparse optimal transport, which maximally aligns the distributions as it can perform conditional transport mappings for each sample (as seen by the differing vertical shifts in (h) depending on where the blue sample lies on the horizontal axis), at the expense of interpretability. Each example shows three levels of decreasing interpretability, where the leftmost column shows the original shift (which has maximal interpretability since $k = 0$) from source (blue diamonds) to target (red down arrows), and the rightmost column shows a shift with near-perfect fidelity.

The second toy example of distribution shift is a shifted Gaussian mixture model which represents a case where groups within a distribution shift in different ways. An example of this type of shift could be explaining the change in immune response data across patients given different forms of treatment for a disease. Looking at (d) in Fig. 4, it is clear that sparse feature transport will not easily explain this shift. Instead, we turn to cluster-based explanations, where we first find $k$ paired clusters and attempt to show how these shift from $P_{src}$ to $P_{tgt}$. Following the mean-shift transport of paired clusters approach outlined in subsection 4.4, the $k = 3$ case as seen in the Appendix shows that three clusters can sufficiently approximate the shift by averaging the shift between similar groups. If a more faithful explanation is required, (f) of Fig. 4 shows that increasing $k$ to 6 clusters can recover the full shift, i.e. PercentExplained=100, at the expense of being less interpretable (which is especially true in a real-world case where the number of dimensions might be large).

The half-moon example, figure (g) in Fig. 4, shows a case where a complex feature-wise dependency change has occurred. This example is likely best explained via feature-wise movement, so will use $k$-sparse transport. If we follow the approach in subsection 4.3 with our interpretable set as the $\Omega^{(k)}$ and let $k = 1$, we get a mapping that is interpretable, but has poor alignment (see Figure (h) in

Fig. 4). For this example, we can possibly reject this explanation due to a poor PercentExplained. With the understanding that this shift is not explainable via just one feature, we can instead use a $k = 2$-sparse OT solution. The $k = 2$ case can be seen in (i) of Fig. 4 which shows that this approach aligns the distributions perfectly, at the expense of interpretability.

## C.2 Explaining Shift in Wisconsin Breast Cancer Dataset

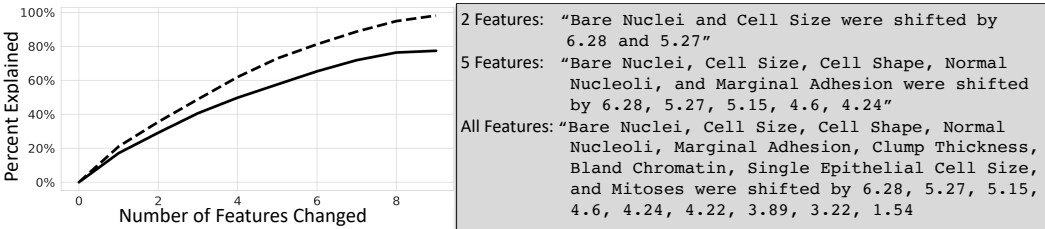

Figure 5: A comparison of the performance of $k$-sparse mean shift explanations (solid line) and $k$-sparse optimal transport explanations (dashed line) when explaining the shift from the benign tumor samples to malignant tumor samples for the UCI Wisconsin Breast Cancer dataset. On the right are example explanations a human operator would see as they change the level of interpretability during $k$-sparse mean shift explanations (where "All Features" is the baseline full mean shift explanation).

This tabular dataset consists of tumor samples collected by Mangasarian & Wolberg (1990) where each sample is described using nine features which are normalized to integers from $[0, 10]$. We split the dataset along the class dimension and set $P_{src}$ to be the 443 benign tumors and $P_{tgt}$ as the 239 malignant samples. To explain the shift, we used two forms of $k$-sparse transport, the first being $k$-sparse mean transport and the second being $k$-sparse optimal transport. The left of Fig. 5 shows that the $k$-sparse mean shift explanation is sufficient for capturing the 50% of the shift between $P_{src}$ and $P_{tgt}$ using only four features, and nearly 80% of the shift with all 9 features. However, if an analyst requires a more faithful mapping, they can use the $k$-sparse OT explanation which can recover the full shift, at the expense of the interpretability. The right of Fig. 5 shows example explanations that an analyst can use along with their context-specific expertise for determining the main differences between the different tumors they are studying.

## C.3 Counterfactual Example Experiment to Explain a Multi-MNIST shift

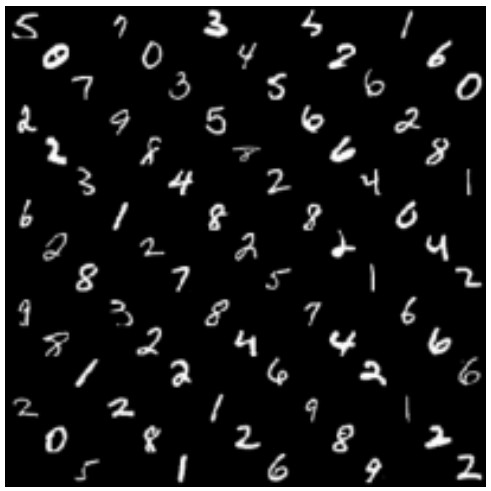 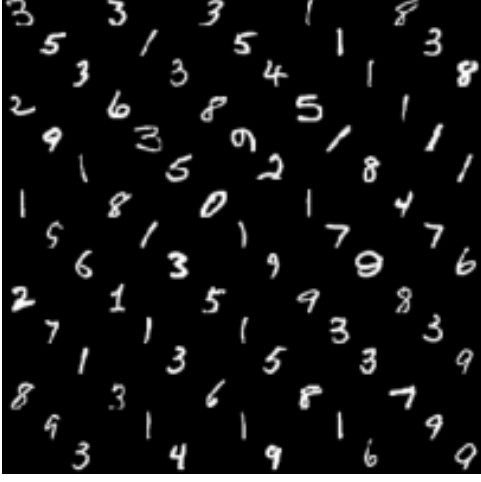

Figure 6: A grid of 25 raw samples from each domain (left is $P_{src}$ and right is $P_{tgt}$). Even for the relatively simple shift seen in the Shifted Multi-MNIST dataset, it may be hard to tell what is different between the two distributions by just looking at samples (without knowing the oracle shift).

As mentioned in subsection 6.2, image-based shifts can be explained by supplying an operator with a set of distributional counterfactual images with the notion that the operator would resolve which semantic features are distribution-specific. Here we provide a toy experiment (as opposed to the real-world experiment seen in subsection 6.2) to illustrate the power of distributional counterfactual examples. To do this, we apply the distributional counterfactual example approach to a Multi-MNIST dataset where each sample consists of a row of three randomly selected MNIST digits Deng (2012) and is split such that $P_{src}$ consists of all samples where the middle digit is even and zero and $P_{tgt}$ is all samples where the middle digit is odd, as seen in Fig. 6.

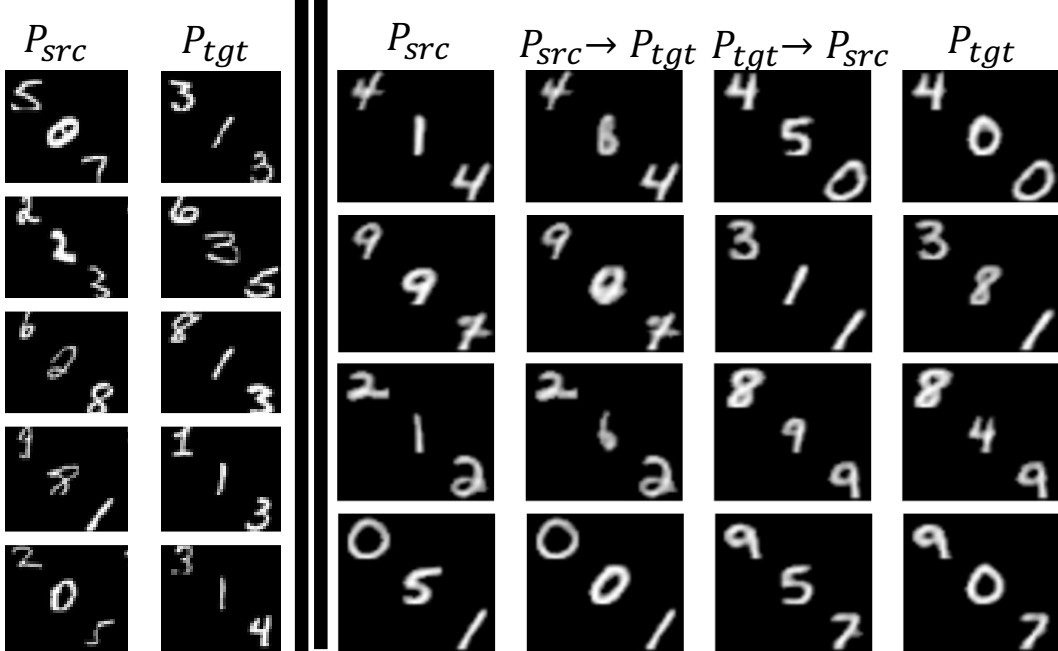

Figure 7: A comparison of the baseline grid of unpaired source and target samples (left) and counterfactual pairs (right) which show how counterfactual examples can highlight the difference between the two distributions. For each image, the top left digit represents the class label, the middle digit represents the distribution label (where $P_{src}$ only contains even digits and zero and $P_{tgt}$ has odd digits), and the bottom right digit is noise information and is randomly chosen. The second, third columns show the counterfactuals from $P_{src} \rightarrow P_{tgt}$ and $P_{tgt} \rightarrow P_{src}$, respectively. Hence we can see under the push forward of each image the "evenness" of the domain digit changes while the class and noise digits remain unchanged.

---

**Algorithm 2** Generating distributional counterfactuals using DIVA

---

**Input:** $\boldsymbol{x}_1 \sim D_1$, $\boldsymbol{x}_2 \sim D_2$, model
$z_{y_1}, z_{d_1}, z_{residual_1} \leftarrow$ model.encode($\boldsymbol{x}_1$)
$z_{y_2}, z_{d_2}, z_{residual_2} \leftarrow$ model.encode($\boldsymbol{x}_2$)
$\hat{\boldsymbol{x}}_{1 \rightarrow 2} \leftarrow$ model.decode($z_{y_1}, z_{d_2}, z_{residual_1}$)
$\hat{\boldsymbol{x}}_{2 \rightarrow 1} \leftarrow$ model.decode($z_{y_2}, z_{d_1}, z_{residual_2}$)
**Output:** $\hat{\boldsymbol{x}}_{1 \rightarrow 2}, \hat{\boldsymbol{x}}_{2 \rightarrow 1}$

---

To generate the counterfactual examples, we use a Domain Invariant Variational Autoencoder (DIVA) Ilse et al. (2020), which is designed to have three independent latent spaces: one for class information, one for domain-specific information (or in this case, distribution-specific information), and one for any residual information. We trained DIVA on the Shifted Multi-MNIST dataset for 600 epochs with a KL-$\beta$ value of 10 and latent dimension of 64 for each of the three sub-spaces. Then, for each image counterfactual, we sampled one image from the source and one image from the target and encoded each image into three latent vectors: $z_y$, $z_d$, and $z_{residual}$. The latent encoding $z_d$ was

then "swapped" between the two encoded images, and the resulting latent vector set was decoded to produce the counterfactual for each image. This process is detailed in Algorithm 2 below. The resulting counterfactuals can be seen in Fig. 7 where the middle digit maps from the source (i.e., odd digits) to the target (i.e., even digits) and vice versa while keeping the other content unchanged (i.e., the top and bottom digits).

### C.4 Using StarGAN to Explain Distribution Shifts in CelebA

Here we apply the distributional counterfactual approach seen in subsection 6.2 to the CelebA dataset Liu et al. (2015), which contains over 200K images of celebrities, each with 40 attribute annotations. We split the original dataset into 5 related sets, $P_1$="blonde hair", $P_2$="brunette hair", $P_3$="wearing hat", $P_4$="bangs", $P_5$="bald". These five sets were chosen as they are related concepts (all related to hair) yet mostly visually distinct. Although there are images with overlapping attributes, such as a blonde/brunette person with bangs, these are rare and naturally occurring, thus they were not excluded.

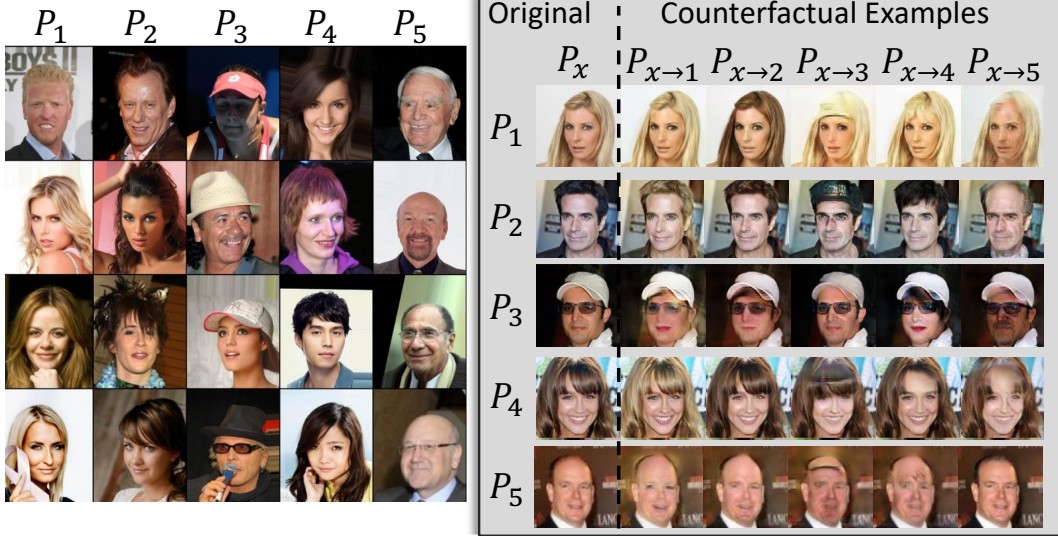

Figure 8: StarGAN is able to adequately translate between distributions with similar content but different style (e.g., $P_1 \rightarrow P_2$), however, when transporting between distributions with different content (e.g., "no hat" $\rightarrow P_3$) the I2I model is unable to properly capture the shift. This is likely due to the model being biased to only change the *style* of the image, while maintaining as much *content* as possible. The figure breakdown is similar to Fig. 3 with the baseline method of unpaired samples on the left and paired counterfactual images on the right, where here $P_1$="blonde hair", $P_2$="brunette hair", $P_3$="wearing hat", $P_4$="bangs", $P_5$="bald".

We trained a StarGAN model Choi et al. (2018) to generate distributional counterfactuals following the same approach seen in subsection 6.2. The result of this process can be seen in Fig. 8, where we can see the model successfully translating "stylistic" parts of the image such as hair color. However, the model is unable to translate between distributions with larger differences in "content" such as removing hair when translating to "bald". This highlights a difference between I2I tasks such as style transfer (where one wishes to bias a model to only change the style of an image while retaining as much of the original content as possible) the mappings required for explaining image-based distribution shifts, which might require large changes in content (such as adding a hat to an image).

## D Explaining Shifts in Images via High-Dimensional Interpretable Transportation Maps

If $x$ is an image with domain $\mathbb{R}^{d \gg 1}$, then any non-trivial transportation map in this space is likely to be hard to optimize for as well as uninterpretable. However, if $P_{src}, P_{tgt}$ can be expressed on some

*interpretable* lower dimensional manifold which is learned by some manifold-invertible function $g : \mathbb{R}^d \rightarrow \mathbb{R}^k$ where $k < d$, we can project $P_{src}, P_{tgt}$ onto this latent space and solve for an interpretable mapping such that it aligns the distributions in the latent space, $P_{T(g(\boldsymbol{x}))} \approx P_{g(\boldsymbol{y})}$. Note, in practice, an encoder-decoder with an interpretable latent space can be used for $g$, however, requiring $g$ to be exactly invertible allows for mathematical simplifications, which we will see later. For explainability purposes, we can use $g^{-1}$ to re-project $T(g(\boldsymbol{x}))$ back to $\mathbb{R}^d$ in order to display the transported image to an operator. With this, we can define our set of high dimensional interpretable transport maps: $\Omega_{\text{high-dim}} := \left\{ T : T = g^{-1}\left(\tilde{T}(g(\boldsymbol{x}))\right), \tilde{T} \in \Omega^{(k)}, g \in \mathcal{I} \right\}$ where $\Omega^{(k)}$ is the set of $k$-interpretable mappings (e.g., $k$-sparse or $k$-cluster maps) and $\mathcal{I}$ is the set of invertible functions with an interpretable (i.e. semantically meaningful) latent space.

Looking at our interpretable transport problem:

$$\underset{T \in \Omega_{\text{high-dim}}}{\arg\min} \; \mathbb{E}_{P_{src}}\left[c(\boldsymbol{x}, T(\boldsymbol{x}))\right] + \lambda_{Fid}\phi(P_{T(\boldsymbol{x})}, P_{\boldsymbol{y}}) \tag{6}$$

Although our transport is now happening in a semantically meaningful space, our transportation cost is still happening in the original raw pixel space. This is undesirable since we want a transport cost that penalizes large semantic movements, even if the true change in the pixel space is small (e.g., a change from "dachshund" to "hot dog" would be a large semantic movement). We can take a similar approach as before and instead calculate our transportation cost in the $g$ space. This logic can similarly be applied to our divergence function $\phi$ (especially if $\phi$ is the Wasserstein distance, since this term can be seen as the residual shift not explained by $T$). Thus, calculating our cost and alignment functions within the latent space gives us:

$$\underset{g \in \mathcal{I}, \tilde{T} \in \Omega^{(k)}}{\arg\min} \; \mathbb{E}_{P_{src}}\left[c\left(g(\boldsymbol{x}), \tilde{T}(g(\boldsymbol{x}))\right)\right] + \lambda\phi(P_{\tilde{T}(g(\boldsymbol{x}))}, P_{g(\boldsymbol{y})}) \tag{7}$$

This formulation has a critical problem however. Since we are jointly learning our representation $g$ and our transport map $T$, a trivial solution for the above minimization is for $g$ to map each point to an arbitrarily small space such that the distance between any two points $c(g(\boldsymbol{x}), g(\boldsymbol{y})) \approx 0$, thus giving us a near zero cost regardless of how "far" we move points. To avoid this, we can use pre-defined image representation function $h$, e.g., the latter layers in Inception V3, and calculate pseudo-distances between transported images in this space. Because $h$ expects an image as an input, we can utilize the invertibility of $g$ and perform our cost calculation as: $c\left(h(\boldsymbol{x}), h\left(g^{-1}\left(\tilde{T}(g(\boldsymbol{x}))\right)\right)\right)$, or more simply, $c_h(\boldsymbol{x}, T(\boldsymbol{x}))$, where $T = g^{-1}\left(\tilde{T}(g(\boldsymbol{x}))\right)$. Similar to the previous equation, we also apply this $h$ pseudo-distance to our divergence function to get $\phi_h$. With this approach, we can still use $g$ to jointly learn a semantic representation which is specific to our source and target domains (unlike $h$ which is trained on images in general, e.g., ImageNet) and an interpretable transport map $\tilde{T}$ within $g$'s latent space. This gives us:

$$\underset{g \in \mathcal{I}, T \in \Omega}{\arg\min} \; \mathbb{E}_{P_{src}}\left[c_h(\boldsymbol{x}, T(\boldsymbol{x}))\right] + \lambda\phi_h(P_{T(\boldsymbol{x})}, P_{\boldsymbol{y}}) \tag{8}$$

While the above equation is an ideal approach, it can be difficult to use standard gradient approaches to optimize over in practice due it being a joint optimization problem and any gradient information having to first pass through $h$ which could be a large neural network. To simplify this, we can optimize $\tilde{T}$ and $g$ separately. With this, we can first find a $g$ which properly encodes our source and target distributions into a semantically meaningful latent space, and then find the best interpretable transport to align the distributions in the fixed latent space. The problem can be even further simplified by setting the pre-trained image representation function $h$ to be equal to the pretrained $g$, since the disjoint learning of $g$ and $\tilde{T}$ removes the shrinking cost problem. By setting $h := g$, we can see that $c\left(h(\boldsymbol{x}), h \circ g^{-1} \circ \tilde{T} \circ g(\boldsymbol{x})\right) = c\left(g(\boldsymbol{x}), \tilde{T} \circ g(\boldsymbol{x})\right) = c_g(\boldsymbol{x}, \tilde{T}(\boldsymbol{x}))$, which simplifies Eqn. 8 back to our interpretable transport problem, Eqn. 6, where $g$ is treated as a pre-processing step on the input images:

$$\underset{T \in \Omega}{\arg\min} \; \mathbb{E}_{P_{src}}\left[c(g(\boldsymbol{x}), g(T(\boldsymbol{x})))\right] + \lambda\phi_g(P_{T(\boldsymbol{x})}, P_{\boldsymbol{y}}) \tag{9}$$

Another way to simplify Eqn. 8 is to relax the constraint that $g$ is manifold-invertible and instead use a pseudo-invertible function such as an encoder $g$ and decoder $g^+$ structure where $g^+$ is a pseudo-inverse to $g$ such that $g^+(g(\boldsymbol{x})) \approx \boldsymbol{x}$. This gives us:

$$\underset{\tilde{T} \in \tilde{\Omega}, g, g^+}{\arg\min} \, \mathbb{E}_{P_{src}} \left[ c_h \left( \boldsymbol{x}, g^+(\tilde{T}(g(\boldsymbol{x}))) \right) \right] + \lambda_{Fid} \, \phi_h(P_{g^+(\tilde{T}(g(\boldsymbol{x})))}, P_{\boldsymbol{y}}) \\ + \lambda_{Recon} \, \mathbb{E}_{\frac{1}{2}P_{src}+\frac{1}{2}P_{tgt}} \left[ L \left( \boldsymbol{x}, g^+(\tilde{T}(g(\boldsymbol{x}))) \right) \right] \tag{10}$$

where $L(\boldsymbol{x}, \cdot)$ is some reconstructive-loss function.

### D.1 EXPLAINING A COLORIZED-MNIST SHIFT VIA HIGH-DIMENSIONAL INTERPRETABLE TRANSPORT

In this section we present a preliminary experiment showing the validity of our framework for explaining high-dimensional shifts. The experiment consists of using $k$-cluster maps to explain a shift in a colorized-version of MNIST, where the source environment is yellow/light red digits with a light grayscale background color (i.e. light gray) and the target environment consists of darker red digits and/or a darker grayscale background colors. Like the lower dimensional experiments before, our goal is to test our method on a shift where the ground truth is known and thus the explanation can validated against. We follow the framework presented in Eqn. 9, where the fixed $g$ is a semi-supervised VAE Siddharth et al. (2017) which is trained on a concatenation of $P_{src}$ and $P_{tgt}$. Our results show that $k$-cluster transport can capture the shift and explain the shift, however, we suspect the given explanation is interpretable because the ground truth is already known. Our hope is that future work will improve upon this framework by better finding a latent space which is interpretable and disentangled, leading to better latent mappings, and thus better high-dimensional shift explanations.

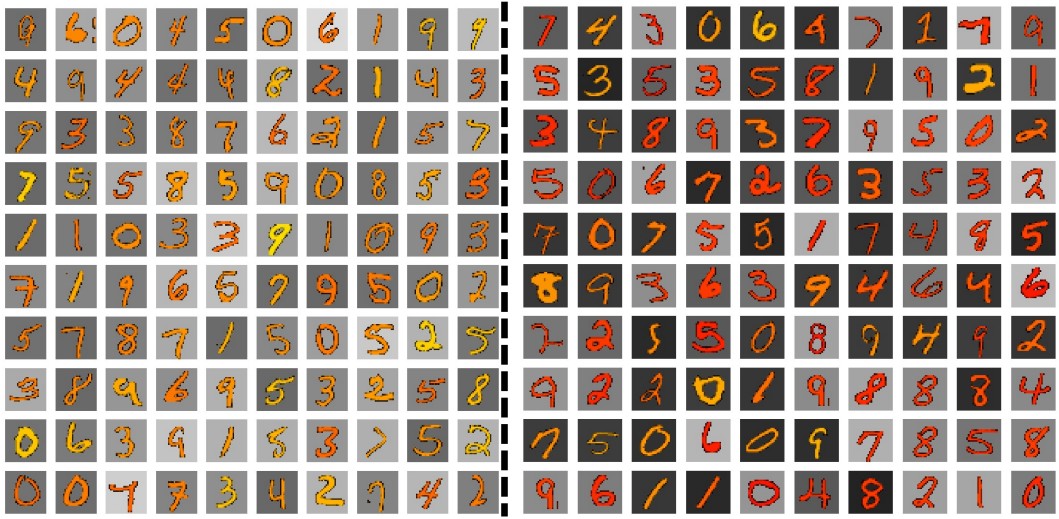

Figure 9: The left figure shows samples from the source environment which has lighter digits and backgrounds while the right figure shows the target environment which has darker digits and/or darker backgrounds

**Data Generation** The base data is the 60,000 grayscale handwritten digits from the MNIST dataset Deng (2012). We first colored each digit by copying itself along the red and green channel axes with an empty blue channel, yielding an initial dataset of yellow digits. We then randomly sampled 60,000 points from a two-dimensional Beta distribution with shape parameters, $\alpha = \beta = 5$. The first dimension of our Beta distribution represented how much of the green channel would be visible per sample meaning low values would result in a red image, while high values would result

in a yellow image. The second dimension of our Beta distribution represented how white vs. black the background of the image would be, where $0 := $ black background and $1 := $ white background.

Specifically, the data was generated as follows. With $x_{raw}$ representing a grayscale digit from the unprocessed MNIST dataset, a mask of representing the background was calculated $\mathbf{m} = x_{raw} \leq 0.1$, where any pixel value below $0.1$ is deemed to be the background (where each pixel value $\in [0, 1]$). Then, the foreground (i.e. digit) color was created $x_{digit-color} = [(1-\mathbf{m}) \cdot x_{raw}, b_1 \cdot (1-\mathbf{m}) \cdot x_{raw}, \mathbf{0}]$, where $\mathbf{0}$ is a zero-valued matrix matching the size of $x_{raw}$ and $b_1 \sim \text{Beta}(\alpha, \beta)$. The background color was calculated via $x_{back-color} = [b_2 \cdot \mathbf{m} \cdot x_{raw}, b_2 \cdot \mathbf{m} \cdot x_{raw}, b_2 \cdot \mathbf{m} \cdot x_{raw}]$. Then $x_{colored} = x_{digit-color} + x_{back-color}$, which results in a colorized MNIST digit with a stochastic foreground and background coloring.

The environments were created by setting the source environment to be any images where $b_1 \geq 0.4$ and $b_2 \geq 0.4$, i.e. any colorized digits that had over 40% of the green channel visible *and* a background at least 40% white, and the target environment is all other images. Informally, this split can be thought of as three sub-shifts: a shift which is only reddens the digit, a second shift which only a darkens the background, and a third shift which is both a digit reddening and background darkening. The environments can be seen in Fig. 9.

**Model**   To encode and decode the colored images, we used a semi-supervised VAE (SSVAE) Siddharth et al. (2017). The SSVAE encoder consisted of an initial linear layer with input size of $3 \cdot 28 \cdot 28$ and a latent size of $1024$. This was then multi-headed into a classification linear layer of size $1024$ to $10$, and for each sample with a label, digit label classification was performed on the output of this layer. The second head of the input layer was sent to a style linear layer of size $1024$ to $50$ which is to represent the style of the digit (and is not used in classification). The decoder followed a typical VAE decoder approach on a concatenation of the classification and style latent dimensions. The SSVAE was trained for 200 epochs on a concatenation of both $P_{src}$ and $P_{tgt}$ with 80% of the labels available per environment, and a batch size of 128 (for training details please see Siddharth et al. (2017)). The transport mapping was then found on the saved lower-dimensional embeddings.

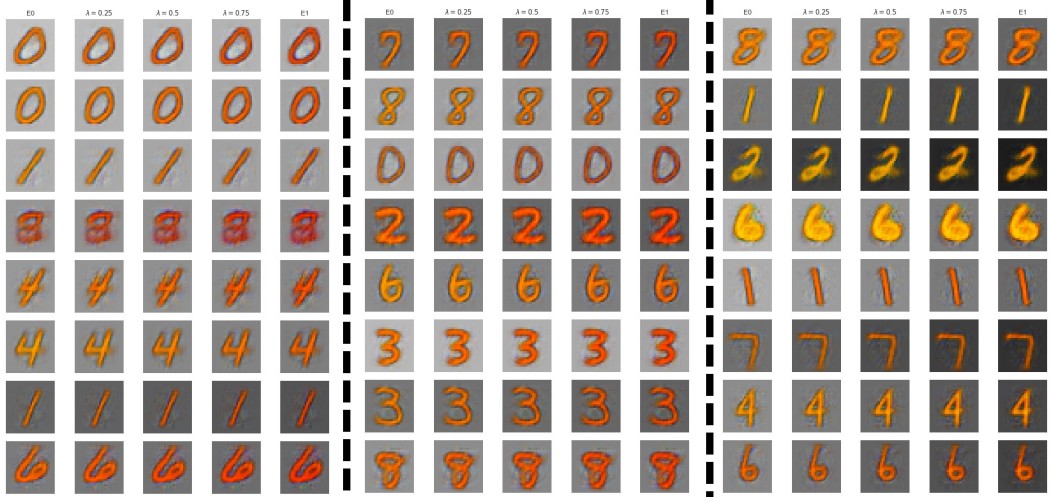

Figure 10: The linear interpolation explanations for the three clusters where the left cluster *seems* to explain the darkening digit shift, the right-most figure explains the shift which darkens the background, and the middle cluster explains the case where both digit and background darkens. For each cluster, the left-most digit $x$ is the reconstruction of original encoding from the source distribution, the right-most digit is the cluster-based push-forward of that digit $T(x)$, and the three middle images are reconstructions of a linear interpolations $\lambda \cdot x + (1 - \lambda) \cdot T(x)$ with $\lambda \in \{0.25, 0.5, 0.75\}$.

**Shift Explanation Results**   Given the shift is divided into three main sub-shifts, we used $k = 3$ cluster maps to explain the shift. We followed the approach given in Eqn. 9, where the three cluster labels and transport were found in the 60 dimensional latent space using the algorithm given in Algorithm 1. Since our current approach is not able to find a latent space with disentangled and

semantically meaningful axes, we cannot use the mean shift information per cluster as the explanation itself (as it is meaningless if the space is uninterpretable). Instead, we provide an operator with $m$ samples from our source environment and the linear interpolation to the samples' push-forward versions under the target environment, for each cluster. The goal is for the operator to discern the meaning of each cluster's mean shift by finding the invariances across the $m$ linear interpolations. The explanations can be seen in Fig. 10.

The linear interpolations from the first cluster (the left of Fig. 10) seem to show a darkening of the source digit, while keeping the background relatively constant. The third cluster (right-most side of the figure) represents the situation where only the background is darkened but the digit is not. Finally, the third cluster seems to explain the sub-shift where both the background and the digit are darkened. However, the changes made in the figures are quite faint, and without *a priori* knowledge of the shift it is possible that this could be an insufficient explanation. As mentioned in Section 6, this could be improved by finding a disentangled latent space with semantically meaningful dimensions, better approximating high dimensional empirical optimal transport maps, jointly finding a representation space and transport map like in Eqn. 4, and more; however, these advancements are out of scope for this work. We hope that this current preliminary experiment showcases the validity of using transportation maps to explain distribution shifts in images and inspires future work to build upon this foundation.

