# OpenReview forum: "Towards Explaining Distribution Shifts"
_ICLR.cc/2023/Conference — Submitted to ICLR 2023_

### Official Review · Reviewer_xS74 · 2022-10-25

**Confidence:** 4
**Correctness:** 3
**Technical Novelty And Significance:** 3
**Empirical Novelty And Significance:** 3
**Recommendation:** 5

**Clarity, Quality, Novelty And Reproducibility:**

The paper is mostly well-organized and written, the proposed methods are also sound and feasible. The explanation of distribution shifts could be important, and this paper proposes transport map based methods, which intend to make the shift interpretable. Extensive evaluation and studies are performed to show the promising of the proposed methods.

**Strength And Weaknesses:**

Strength:

- Important problem
- Sound and feasible solution
- Extensive evaluation and showcase to demonstrate the potential usefulness

Weakness:

- The selection of hyperparameter is somehow empirical, lacks constructive ways to obtain a most optimized k.
- Not very clear how to deal with cases that do not have balanced samples from source and target data distribution.
- Not clear whether the obtained k-sparse and k-sparse cluster always have real and interpretable meanings.
- Lack of detailed discussion on how the obtained explanation of the shifts could be useful in practice, in an actionable way for an operator.

**Summary Of The Paper:**

This paper proposes an interpretable transport map based method for explaining the distribution shift. It further proposes methods to calculate and obtain k-spare and k-cluster transport, based on pre-defined hyperparameters. Extensive evaluation has been performed to demonstrate the promising of the proposed methods.

**Summary Of The Review:**

Overall, I think this paper works on an important problem with a feasible solution.

Although the proposed transport map based method seems to be simple, less well-grounded, and somehow heuristic, I think it could be a useful step and feel positive about this paper. I also appreciate the authors’ effort in extensive evaluation and studies. Even though, I hope the authors could address the concerns, posted in the weakness part, during the response.

After rebuttal:
I appreciate the authors' feedback on the posted questions. However, at the current stage, it would be very important to make the proposed method actionable, since I would say it is not at a very early stage. Even though, I do appreciate the authors' effort in working in this direction and it is important, and hope authors could address this critical issue and enhance the paper in the future version.

---

> ### Author Response · Authors · 2022-11-17
> **Response to Reviewer xS74 [part 1 of 2]**
>
> Thank you for your review and for recognizing the significance of explaining distribution shifts and the comprehensiveness of our experiments. As you mentioned, the explanation of distribution shifts could be important --due to the ubiquity of the problem (especially in real-world systems), we hope that upon acceptance this work will excite researchers from closely related fields (e.g., data summarization/understanding, explainable AI, shift/anomaly detection) to begin advancing this new area. We hope our response highlights the actions that can be taken in response to distribution shift explanations, and if you have any additional questions/concerns, we will be happy to answer any questions during the discussion period.
>
> > The selection of hyperparameter is somehow empirical, lacks constructive ways to obtain a most optimized k.
>
> The selection of the $k$ hyperparameter (i.e. the level of interpretability of the explanation) is an operator-specific process, and thus we optimize its selection in general--as different operators+settings might require more precise (and thus less interpretable) mappings or more easily understood mappings (at the expense of alignment accuracy). The rule of thumb we follow in the paper is to set a target PercentExplained value, examine the explanation for the corresponding $k$ value, and adjust the interpretability from there as needed via increasing the PercentExplained of the mapping by increasing $k$ or making the explanation easier to interpret via using fewer cluster or shifting fewer features (i.e. decreasing $k$).
>
> > Not very clear how to deal with cases that do not have balanced samples from source and target data distribution.
>
> This is not a problematic situation for our method allows for using entropic optimal transport [1] which can find an OT solution with unbalanced data distributions. For the case of $k$-cluster mappings, we can employ a similar entropic pairing method to allow the splitting of mass of samples between clusters in the source and target distributions (i.e. one-to-many cluster mappings).
>
> > Not clear whether the obtained k-sparse and k-sparse cluster always have real and interpretable meanings.
>
> We discuss this limitation in Appendix B, where we agree it is not always guaranteed that $k$-sparse and $k$-cluster explanations find semantically meaningful mappings. For example, performing $k$-cluster transport in a setting where no natural clusters exist, e.g., $P_{src} \sim $Uniform$(0, 1)$ and $P_{tgt} \sim $Uniform$(1, 2)$, then the resulting cluster transports are likely to be meaningless (although using $k$-sparse mappings would be able to recover the shift). While this cannot be avoided in general, using a clustering method that is either specifically designed for finding interpretable clusters [2, 3] or one which directly optimizes the objective in interpretable transport equation might lead to explanations that are easier to interpret or validate.

---

> > ### Author Response · Authors · 2022-11-17
> > **Response to Reviewer xS74 [part 2 of 2]**
> >
> > > Lack of detailed discussion on how the obtained explanation of the shifts could be useful in practice, in an actionable way for an operator.
> >
> > While actionable insights have to be posed, they are for very specific contexts. We note much of the explainability literature for predictions still lacks this specific actionable insight (especially early works). We definitely think this is an important next question to ask and we aim to give examples of actionable takeaways for each experiment in the paper. For example, the second paragraph in the adult income experiment is a hypothetical case study of a researcher looking to implement a social program targeting gender inequalities which, armed with our $k$-sparse explanation, leads the practitioner to investigate the discrepancy between bachelor's educated men and women.
> >
> > Additional actionable explanations include $k$-cluster explanations of how heterogenous groups of consumers reacted differently to an advertising campaign (where $P_{src}$ is user behaviors before seeing the ad and $P_{tgt}$ post-impression behaviors). This can help identify which subpopulations are interested in the product. Actions from $k$-sparse shifts can come from learning how specific features have changed in an operating environment. For example, when debugging a shift in a sensor network’s distribution, learning the shift was caused by a sensor in the network acting erroneously allows one to fix the problem--possibly without needing to resort to an expensive ML retraining process (e.g., simply replacing the sensor would solve the problem).
> >
> >
> > [1] Peyré, Gabriel, and Marco Cuturi. "Computational optimal transport: With applications to data science." Foundations and Trends® in Machine Learning 11.5-6 (2019): 355-607.
> >
> > [2] Fraiman, Ricardo, Badih Ghattas, and Marcela Svarc. "Interpretable clustering using unsupervised binary trees." Advances in Data Analysis and Classification 7.2 (2013): 125-145.
> >
> > [3] Bertsimas, Dimitris, Agni Orfanoudaki, and Holly Wiberg. "Interpretable clustering: an optimization approach." Machine Learning 110.1 (2021): 89-138.

---

> ### Author Response · Authors · 2022-11-28
> **Have we addressed your questions?**
>
> Hi Reviewer xS74, thank you again for your helpful feedback. Have we adequately addressed your questions and concerns?
>
> In addition to reinforcing the importance of gaining a better understanding of distribution shifts and the feasibility of our solution, we hope
> to have clarified how our approach can yield actionable takeaways such as using $k$-cluster maps to see if a policy change positively affected minority subpopulations in the intended manner. We also hope to have shown how one should adjust $k$ to yield the best explanation for a given context as well as when to use $k$-sparse vs. $k$-cluster methods.
>
> Please let us know if we can answer any more questions or doubts, as we will be happy to answer them. Thank you!

---

### Official Review · Reviewer_Sirc · 2022-10-25

**Confidence:** 4
**Correctness:** 3
**Technical Novelty And Significance:** 3
**Empirical Novelty And Significance:** 3
**Recommendation:** 5

**Clarity, Quality, Novelty And Reproducibility:**

This paper is clearly demonstrated. The quality and novelty are not enough in terms of significance and technical contributions. This paper can be reproducible based on the algorithm provided.

**Strength And Weaknesses:**

Pros:

1.  This paper contributes an interesting idea to see where the distribution changes.

2. This paper is clear and easy to follow.


Cons:

1. The major problem contained in this paper is that the criterion design is not well motivated. Eq. (2) does not have a clear motivation.

2. Discussions regarding the down-stream tasks are not enough. It is hard to believe that understanding the distribution shift can be directly applied in some scenarios.

3. The motivations behind the K-S and K-C are not clear as well.

4. How to apply the results to improve the general machine learning algorithm? Distribution shifts happen everywhere, what we care about is how to adapt to the distribution shift. After the understanding part, it is very important to say how to use this kind of results.

5. Adversarial data-based distribution shift might be a good example to analyze.


**Summary Of The Paper:**

In this paper, the authors introduced a framework for explaining distribution shift using a transport map T between a source and target distribution. They constrained a relaxed form of optimal transport to theoretically define an interpretable mapping TIT and introduced two interpretable transport methods: k-sparse and k-cluster transport. The authors also provided practical approaches to calculating a shift explanation, which allows us to use treat interpretability as a hyperparameter that can be adjusted based on a user’s need and showed how our methods can help an operator investigate a distribution shift on real-world examples.

**Summary Of The Review:**

The motivations in this paper are not clear. In addition, it is hard to see the potential application of the results when there are no extensive discussions regarding the distribution shifts considered in the machine learning community.

---

> ### Author Response · Authors · 2022-11-17
> **Response to Reviewer Sirc [part 1 of 2]**
>
> Thank you for your detailed review and for recognizing the value of our approach! Since distribution shift is such a ubiquitous problem (especially in real-world systems), we hope that our response can clear any confusion with our motivation. If you have any additional questions/concerns, we will be happy to answer any questions during the discussion period.
>
> > The major problem contained in this paper is that the criterion design is not well motivated. Eq. (2) does not have a clear motivation.
>
> We are unsure of what you mean by “criterion design”, but we will assume this means “how we evaluate the explanations” (please correct us if this assumption is wrong, and we’ll happily respond to the true meaning). As mentioned in our response to Reviewer 601f, it is impossible to measure the efficacy of an explanation in general (since the effectiveness is dependent on the operator and the setting--approximating interpretability and actionability of an explanation is a highly active area of research [1] [2]). Since we cannot directly measure the effectiveness of an explanation, we instead looked at what we can measure about an explanation. Towards this end we decided we can measure/approximate: 1) the relative interpretability, hence why we focus on $k$-level-explanations--since $k$ allows us to control the relative interpretability (see section 4.1) and (2) how accurate the explanation is, which we operationalize via the PercentExplained metric (Eq. 2). The PercentExplained shows how well a given explanation $T$ aligns the source and target distributions. We note it is not a measurement of the full explainability of $T$, but rather a measurement of how well $T$ “fits” the distribution shift--similar to how an $R^2$ value shows the fit of a regression model (we show this correspondence in the paragraph after Eq. 2).  While the PercentExplained metric alone might not be sufficient (similar to how an $R^2$ value is not sufficient in showing the usability of a regression model), our claim is that given the PercentExplained, an adjustable $k$ value, and a corresponding $T_{IT}$ explanation, an operator will be better equipped to understand a distribution shift than if they were to use a different approach such as a mean-shift explanation or Shapley values for a distribution classifier.
>
>
>
> > It is hard to believe that understanding the distribution shift can be directly applied in some scenarios.
>
> We agree that explaining distribution shifts can be quite tricky (or impossible) in some scenarios (similar to explaining deep machine learning models, which, despite this, is still a rich area of research [2]). As mentioned in our Challenges and Limitations section (Appendix B), a common case is when the original data is uninterpretable. This is problematic since having a baseline understanding of the system/data is a prerequisite for explainability [1]. Uninterpretable data typically can take two forms: 1) the raw data features are uninterpretable but the samples are interpretable (e.g., images are interpretable but the pixel-level features are not) or 2) when both the raw data features and samples are uninterpretable (e.g., raw outputs from material science simulations). For (1), using sets of input-output distributional counterfactual pairs is an effective explanation scheme (as laid out in Section 6.2 and showcased in Figure 3.). The second setting (2), is more difficult and requires finding an interpretable latent feature space (e.g., semi-causal mechanisms for the material science simulation) in which one can ground their explanations. These are problems not limited to our method but rather a characteristic of distribution shifts in general. Although many distribution shifts might not fall under the settings described in (1) and (2), due to the ubiquity and the possibly severely problematic nature of distribution shifts [3, 4], each of these areas are important to advance. Our goal in this work is to initiate that effort.

---

> > ### Author Response · Authors · 2022-11-17
> > **Response to Reviewer Sirc [part 2 of 2]**
> >
> > > The motivations behind the K-S and K-C are not clear as well.
> >
> > We apologize for being unclear here. Our overall goal is to find mappings that can align the source and target distributions in an interpretable way (e.g., how mean shift shows how the averages of the source and target distributions differ). To this end, we introduce two general mappings: the $k$-sparse and $k$-cluster mappings. (where, as mentioned above, the motivation for $k$ is to allow for relative measurements of interpretability). The motivation for the $k$-sparse mappings is to allow for a transport map to only operate on a subset of the features, since trying to gain a *deep* understanding of how a change has happened across 10 dimensions is much harder than understanding how a change has happened across only 2 dimensions (e.g., explaining the shift between benign tumor and malignant tumor samples in the Wisconsin Breast Cancer dataset experiment in Figure 5.). Thus the $k$-sparse mappings are useful to allow an operator to see how specific *features* changed from $P_{src}$ to $P_{tgt}$.
> >
> > On the other hand, the $k$-cluster mappings allow an operator to track how sub-groups within $P_{src}$ changed under the distribution shift. This can be seen in the Adult Income experiment (Figure 2.), where the fourth cluster $C^4$ highlights a group that has a much larger income gap between males ($P_{src}$) and females ($P_{tgt}$). More generally, it is easy to imagine how $k$-cluster mappings can be used to explain how heterogenous groups of consumers reacted differently to an advertising campaign (where $P_{src}$ is their behavior before seeing the ad and $P_{tgt}$ is post-impression behavior). Thus, $k$-cluster mappings assist an operator to see how specific *groups of samples* changed under the distribution shift.
> >
> > > How to apply the results to improve the general machine learning algorithm? Distribution shifts happen everywhere, what we care about is how to adapt to the distribution shift. After the understanding part, it is very important to say how to use this kind of result.
> >
> > While adapting to a distribution shift is important (and a very active area of research [5]), ML-related problems caused by distribution shifts can be fixed by methods beyond adaption/retraining. Namely, a distribution shift can typically be a signal that an operating environment has changed from what a model was originally changed in, thus, discovering what is different (e.g., a sensor in a network has failed) can be crucial to fixing the problem--possibly without needing to resort to an expensive ML retraining process (e.g., simply replacing the sensor would solve the problem). This also can be used to determine what types of robustness need to be built into an ML model (e.g., implementing a sensor dropout augmentation into one’s training setup). More generally, distribution shifts are not limited to ML problems, and understanding them are important in many other contexts, such as knowledge discovery via tracking consumer preferences, as mentioned above.
> >
> > > it is hard to see the potential application of the results when there are no extensive discussions regarding the distribution shifts considered in the machine learning community.
> >
> > We would like to note our experiments involve explaining multiple distribution shifts from the Stanford WILDS database [6] which is a highly acclaimed benchmark set of real-world distribution shifts. As with deep learning explanations, this is a novel area and we only claim to take one useful step along this direction. We hope our work inspires more in-depth exploration of shift explanations (similar to what has happened for general explainability/interpretability of predictions).
> >
> >
> > [1] Molnar, Christoph. Interpretable machine learning. Lulu. com, 2020.
> >
> > [2] Lipton, Zachary C. "The mythos of model interpretability: In machine learning, the concept of interpretability is both important and slippery." Queue 16.3 (2018): 31-57.
> >
> > [3] Beery, Sara, Grant Van Horn, and Pietro Perona. "Recognition in terra incognita." Proceedings of the European conference on computer vision (ECCV). 2018.
> >
> > [4] Rabanser, Stephan, Stephan Günnemann, and Zachary Lipton. "Failing loudly: An empirical study of methods for detecting dataset shift." Advances in Neural Information Processing Systems 32 (2019).
> >
> > [5] Wang, Mei, and Weihong Deng. "Deep visual domain adaptation: A survey." Neurocomputing 312 (2018): 135-153.
> >
> > [6] Koh, Pang Wei, et al. "Wilds: A benchmark of in-the-wild distribution shifts." International Conference on Machine Learning. PMLR, 2021.

---

> ### Author Response · Authors · 2022-11-28
> **Have we addressed your questions?**
>
> Hi Reviewer Sirc, thank you again for your helpful feedback. Have we adequately addressed your questions and concerns?
>
> We hope to have clarified the motivation for explaining distribution shifts as they are ubiquitous and can lead to serious consequences if not properly handled/debugged. With this, we then aimed to show how an (adjustable) interpretable transportation map (e.g., $k$-cluster maps) and PercentExplained can allow an operator to understand a distribution shift better than if they were to use de facto standard approaches such as a mean-shift explanation or Shapley values for a distribution classifier.
>
> If you still have doubts regarding these topics (or any other), please let us know and we will be happy to discuss them. Thank you!

---

> ### Author Response · Authors · 2022-12-09
> **Author Response Follow-Up**
>
> Hi Reviewer Sirc, given our response and updated paper which should make the motivation clear and highlight the utility of explaining distribution shifts, would you consider raising your score?
>
> Thank you again for your review and your time.

---

### Official Review · Reviewer_6o1f · 2022-10-26

**Confidence:** 3
**Correctness:** 3
**Technical Novelty And Significance:** 3
**Empirical Novelty And Significance:** 2
**Recommendation:** 5

**Clarity, Quality, Novelty And Reproducibility:**

The clarity of the paper can be improved in several ways (see above). The methods proposed are original to the best of my knowledge.

**Strength And Weaknesses:**

Strengths:
- The paper tackles an important and practical problem in distribution shifts.
- The authors test their method on a variety of datasets from different modalities, including images.

Weaknesses:
1. There are several components of the method which are not explained very clearly in the paper:
- It is unclear how the argmin problem in Equation 3 is actually solved. Is it always possible to decompose this function into a sum over individual features? What is the computational complexity of this problem?
- I cannot find where the authors state the form of the $c$ function.
- It is not clear what types of distribution shifts the proposed methods can identify. It seems that the OT method can identify arbitrary conditional shifts (i.e. $P(X_i | X_{\neg i})$), whereas the mean method can only pick up marginal shifts ($P(X_i)$). The authors should clarify this, and perhaps contrast the difference with a new synthetic dataset in the appendix.
- Is the output of the method dependent on the relative scale of the features? It seems that the authors normalize the Adult dataset, but do not state this fact explicitly.
- The experiment in Section 6 is missing some details. For example, what is $\Omega$ defined in $\Omega_{high-dim}$ at the top of page 8? If it is a set of k-sparse mappings, then what is $k$ in Figure 3?

2. The authors propose the PercentExplained metric, which is a measure of the effectiveness of the _mapping_. However, in practice, we only care about the effectiveness of the _explanation_. As an example, in the CivilComments result from Table 1, it seems that the output of the mean method (in terms of the features selected) is very similar to that of the OT method. For example, the top 4 features are identical for the F->M case. It seems that the reason why the OT method has much higher PercentExplained is because its underlying mapping is better. To compare the explanations of the two methods fairly, the authors should consider showing the PercentExplained for the mean method using the underlying mappings from OT.

3. The output of the method for the Adult Income experiment in Figure 2 does not seem very meaningful to me. For example, the first cluster $C_M^1$ (presumably accounting for 80% of the shift), when mapped to the target (female) distribution, lowers the average age and the average education level, with a small increase in the average income. Does this indicate that middle-aged men have similar income to women who are slightly less educated and younger? It is not clear to me how this result would be actionable to a practitioner.

4. The authors only compare against very simple baselines. For the Section 6 experiment, it seems that many I2I methods (which the authors reference) such as CycleGAN would be reasonable baselines. For the Section 5 experiment, I think Section 3 from Budhathoki et al., 2021 could work, using the attribution magnitude as a ranking.

5. There are several spelling mistakes in the paper. For example, "yeild" at the top of page 4, a missing bracket in Equation 3, $P_y$ which should probably say $P_{tgt}$ at the end of Section 4.2, and a left over comment in Appendix D.

6. Overall, the proposed methods do not seem very cohesive, and instead seems like a patchwork of approaches for different settings.

**Summary Of The Paper:**

The authors explore explaining distribution shifts using interpretable transport mappings. The authors first propose a method which identifies $k$ features which maximally account for the optimal transport solution. They evaluate this method on the CivilComments dataset, finding that it outperforms a mean shift baseline. Then, the authors propose a method for explaining shifts by selecting $k$ clusters, which they evaluate on the Adult Income dataset. Finally, the authors demonstrate their ability to explain shifts in images by applying interpretable transport mappings to the latent space of a generative model to generate counterfactual images.

**Summary Of The Review:**

The paper presents an interesting and novel method for explaining distribution shifts using transport mappings. However, I believe that several aspects of the method are not well explained or are not well explored, and the empirical results are not particularly convincing. I am leaning towards rejection pending the rebuttal.

---

> ### Author Response · Authors · 2022-11-17
> **Response to Reviewer 601f [part 1 of 2]**
>
> Thank you for your thorough review and for acknowledging the strengths of this work (e.g., "important", “novel”, and “practical”) as well as areas that need work. While we tried to address most weaknesses/areas we were unclear, if you have any additional questions/concerns, we will be happy to answer any questions during the discussion period.
>
> > It is unclear how the argmin problem in Equation 3 is actually solved. Is it always possible to decompose this function into a sum over individual features?
>
> The argmin problem in Eq 3. can be solved using standard constrained optimization methods to search over the given $\Omega$ class or one instead optimize an unconstrained variant using Lagrangian relaxation (e.g., penalizing any $T$ which moves more than $k$ features in total). The $k$ explanation methods we use in this work minimize this objective for the unconstrained $\Omega^{(k)}$ and $\Omega^{(k)}_{vector}$ exactly (see proofs A.4 and A.5) and thus we do not need to run this optimization. Additionally, we note the summation in Eq 3. is over *samples* rather than over individual features.
>
>
> > It is not clear what types of distribution shifts the proposed methods can identify. It seems that the OT method can identify arbitrary conditional shifts (i.e. $P(X_i|X_{-i})$ ), whereas the mean method can only pick up marginal shifts (P(Xi)). The authors should clarify this, and perhaps contrast the difference with a new synthetic dataset in the appendix.
>
> It is correct that the $k$-OT method can align $P_{src}$ and $P_{tgt}$ using arbitrary conditional mappings (and if $k=n_{features}$, then $k$-OT will perfectly align $P_{src}$ and $P_{tgt}$). However, we would like to note that $k$-$\mu$ cannot perform arbitrary marginal transports, but rather it can only add a scalar value to each dimension in the active feature set $\mathcal{A}$  (i.e. $T_{k-\mu}(\mathbf{x}) = \mathbf{x} + \tilde{\mathbf{\delta}}$).  We originally mentioned this in section 4.3 and tried to highlight its effect in the CivilComments experiment (Table 1.), but thanks to your suggestion have made this more clear in the revised paper (in section 4.3 and the synthetic experiments in Appendix C.1)
>
> > Is the output of the method dependent on the relative scale of the features? It seems that the authors normalize the Adult dataset, but do not state this fact explicitly.
>
> Yes, the output of the method is dependent on the relative scale of the features (similar to PCA). While this can easily be fixed by standardizing the dataset (as you mentioned), which we do in the Adult Income experiment (mentioned halfway into the first paragraph), this should only be done if one wants an explanation that ignores the scaling of units.
>
> > The experiment in Section 6 is missing some details. For example, what is $\Omega$ defined in $\Omega_{high−dim}$ at the top of page 8? If it is a set of k-sparse mappings, then what is k in Figure 3?
>
> We apologize for the confusion here.  When defining $\Omega_{high−dim}$, we claim $\Omega$ is the set of interpretable mappings--referring to the original $\Omega$ seen in Eq. (1) (e.g., $k$-sparse or $k$-cluster mappings).  While $\Omega_{high−dim}$ is discussed in detail in Appendix D (due to space constraints), we have added this clarification in Section 6 in the revised paper. As for Figure 3., this experiment does not use the method described in Section 6.1 ($T_{HIT}$) but rather uses a StarGAN model to generate distributional counterfactuals as mentioned in Section 6.2.

---

> > ### Author Response · Authors · 2022-11-17
> > **Response to Reviewer 601f [part 2 of 2]**
> >
> > > The authors propose the PercentExplained metric, which is a measure of the effectiveness of the mapping. However, in practice, we only care about the effectiveness of the explanation
> >
> > This is a very good point, and we thank you for bringing this up. While it would be ideal to have a metric that shows the effectiveness of a given mapping, this is impossible to measure in general (since the effectiveness is dependent on the operator and the setting--approximating interpretability and actionability of an explanation is a highly active area of research [1] [2]), and thus we can only approximate this desideratum. Since we cannot directly measure the effectiveness of an explanation, we instead looked at what we *can* measure about an explanation (similar to [3] which defines quantifiable metrics for feature attribution methods). Towards this end we decided we can measure/approximate: 1) the relative interpretability, hence why we focus on $k$-level-explanations--since $k$ allows us to control the relative interpretability (see section 4.1) and (2) the accuracy/fidelity of the explanation, this being measured via PercentExplained (similar to [3] or to how a $R^2$ value shows the fit of a regression model). Indeed, the PercentExplained metric does not directly measure the explanation effectiveness, it does tell the user what is *not* being explained, i.e., the known unknowns. While the PercentExplained metric alone might not be sufficient, our claim is that given the PercentExplained, an **adjustable** $k$ value, and a corresponding $T_{IT}$ explanation, an operator will be better equipped to understand a distribution shift than if they were to use a different approach such as a mean-shift explanation or Shapley values for a distribution classifier.
> >
> >
> > > [In the Adult Income experiment,] the first cluster $C_M^1$ (presumably accounting for 80% of the shift), when mapped to the target (female) distribution, lowers the average age and the average education level, with a small increase in the average income. Does this indicate that middle-aged men have similar income to women who are slightly less educated and younger? It is not clear to me how this result would be actionable to a practitioner.
> >
> > To clarify, the x-axis in the plot in the bottom left of Figure 2. refers to the *total* number of clusters for a $k$-cluster explanation. So the $k$=1 case you mention (which explains ~80% of the shift) is equivalent to the mean-shift case, which can be seen in the top left of the figure. While does have some actionable insights (e.g., the slight drop in income from M->F), it is an average over a full dataset leading to a 20% level of error, which we agree does not seem actionable. However, in our method, the 4-cluster explanation, we can see this explains the shift with ~90% accuracy and in $C^4$ highlights the quite large income disparity in middle-aged adults with a bachelor's degree from males having nearly a 100% likelihood of having an income $\geq$50k to only a 38% chance when pushed onto the female cluster.
> >
> > > The authors only compare against very simple baselines. For the Section 6 experiment, it seems that many I2I methods (which the authors reference) such as CycleGAN would be reasonable baselines. For the Section 5 experiment, I think Section 3 from Budhathoki et al., 2021 could work, using the attribution magnitude as a ranking.
> >
> > As mentioned in our general author response, since this is a new area that has no published baselines. Thus, we have to compare against what baselines feel natural to the problem (e.g., training a distribution classifier and using feature attribution methods to explain the shift via features). Regarding the experiment in section 6, this currently uses the I2I method, StarGAN to produce the distribution counterfactuals, but indeed other I2I methods could be used. For the section 3 experiment, Budhathoki et al., 2021 requires knowledge of a causal graphical model, which we do not assume and would require complex causal graphical model learning, which is out of scope for our current paper.
> >
> > > Paper cohesiveness
> >
> > We apologize for any confusion when reading the paper. We have gone through and made sure to fix all mistakes we could find. Hopefully, this will aid in the overall comprehension of the paper, and we hope this is not a rejection-worthy issue (we also note that other reviewers mentioned the paper was well-written, clear, and easy to follow).
> >
> > [1] Molnar, Christoph. Interpretable machine learning. Lulu. com, 2020.
> >
> > [2] Lipton, Zachary C. "The mythos of model interpretability: In machine learning, the concept of interpretability is both important and slippery." Queue 16.3 (2018): 31-57.
> >
> > [3] Yeh, Chih-Kuan, et al. "On the (in) fidelity and sensitivity of explanations." Advances in Neural Information Processing Systems 32 (2019).

---

> ### Author Response · Authors · 2022-11-28
> **Have we addressed your questions?**
>
> Hi Reviewer 6o1f, thank you again for your thorough and helpful feedback. Have we adequately addressed your questions and concerns?
>
> In addition to reinforcing the ubiquity and importance of gaining a better understanding of distribution shifts, we hope to have made the paper more cohesive (e.g., how and when to use $k$-sparse shifts vs. $k$-cluster shifts,  the importance of the PercentExplained metric, clarified experiments and their actionable takeaways). We also hope to have shown that the evaluations of our method are extensive (5 experiments on real-world data + 5 experiments on simulated data), especially given the context that this work is in a new field (explaining distribution shifts) which has no formal benchmarks to evaluate against as well as this method being a general method in explainability--which is already hard to define and evaluate.
>
> Please let us know if we can answer any more questions, as we will be happy to answer any. Thank you!

---

> ### Author Response · Authors · 2022-12-09
> **Author Response Follow-Up**
>
> Hi Reviewer 6o1f, given our response and updated paper which should fix cohesivity concerns and the further details of our experimental evaluations (e.g., the importance of the PercentExplained metric and the lack of official baselines due to a new field), would you consider raising your score?
>
> Thank you again for your review and your time.

---

### Official Review · Reviewer_Lnku · 2022-11-04

**Confidence:** 3
**Correctness:** 4
**Technical Novelty And Significance:** 2
**Empirical Novelty And Significance:** 2
**Recommendation:** 5

**Clarity, Quality, Novelty And Reproducibility:**

Clarity: The paper is well written
Quality: The idea builds on optimal transport and the authors introduce elements to make it feasible for the purpose of interpretability.
Reproducibility: The work provides details to reproduce the work

**Strength And Weaknesses:**

+  Important research problem
+  Interesting idea of building on OT and relaxing the problem to make it feasible
+  Realistic evaluation

-  Excessive reliance on domain expertise
-  Lack of qualitative evaluation

**Summary Of The Paper:**

The authors focus on providing an approach to interpret distributions
shifts. To this end, they build on Optimal Transport by considering a
relaxation of the problem. They propose two general ways to approach
interpretability of distribution shifts that trade off complexity for
interpretability. Results on real-world data show the approach is
promising in providing explanation of distribution shifts.

**Summary Of The Review:**

The idea of approaching the interpretability of distribution shifts with
transport maps is interesting; the relaxation from optimal to
interpretable transport helps narrow OT to a set of interpretable OT
\Omega. One concern revolves around the need of relying excessively on
human knowledge to define \Omega; I do understand the problem is
challenging and the authors propose two general methods to kickstart the
process. It would be great to integrate this reasoning in the
experimental section too.

A key point of achieving IT, however, is to rely on k-Sparse Transports.
how can we identify what k dimensions to transport points from P_src to
P_tgt? Does this also need to be user-defined? Similar reasoning applies
to k-Cluster Transport. It seems the approach requires domain knowledge
and this is clearly applicable in specific contexts, where dimensions
and clusters are still manageable, for instance, as the experiments show.

The evaluation is interesting although I am not sure the benefit of the
authors' approach stands out. Looking at the k-mean explanation only on
the specific evaluation shown in Table 1, it is possible to understand
the shift quite well without resorting to the authors' k-OT approach.
The authors provide an explanation in Section 5 but it still seems
blurred.  Is it because of the higher percentage? That comes with a
higher computation induced by the authors' approach. What is missing
here is a qualitative evaluation that would help humans to understand
what approach helps the most. (Also, how would this compare against
Shapley values?)

---

> ### Author Response · Authors · 2022-11-17
> **Response to Reviewer Lnku**
>
> Thank you for your review and for recognizing the promise and importance approach! While we tried to address all comments, if you have any additional questions/concerns, we will be happy to answer any questions during the discussion period.
>
> > One concern revolves around the need of relying excessively on human knowledge to define $\Omega$
>
> We agree defining an interpretable set can be difficult to do. Thus, we dedicate much of our paper to defining the $k$-sparse, $k$-cluster, and distributional counterfactual explanation approaches which we believe to be general and can be applied to many situations involving distribution shifts--which we validate in our multiple experiments on diverse real-world cases. However, as mentioned in our limitations section and in our general response to reviewers, we believe defining new $\Omega$ sets which are useful for specific contexts is a ripe area for future advancements.
>
> > how can we identify what k dimensions to transport points from $P_{src}$ to $P_{tgt}$? Does this also need to be user-defined?
>
> Apologies for the confusion. While which dimensions are used in $k$-sparse transport (i.e. we refer to this as the active feature set, $\mathcal{A}$) *can* be defined by a user, in our approach, we set  $\mathcal{A}$ to be the $k$ features which are most different from $P_{src}$ and $P_{tgt}$. In other words, these are the $k$ features which, if included in  $\mathcal{A}$, will align the two distributions the most, as defined by the $W^2_2$ distance between $P_{src}$ and $P_{tgt}$. This is explained in detail in section 4.3, and we are happy to answer any additional questions.
>
> > Similar reasoning applies to k-Cluster Transport [how to select the cluster partitions]. It seems the approach requires domain knowledge and this is clearly applicable in specific contexts.
>
> Similar to that of the $k$-sparse transport maps, under our solution the finding of the $k$ clusters is automatic. As mentioned in section 4.4 and Algorithm 1, the clustering is performed in a joint space between P_src and P_{T_{OT}(\mathbf{x}}}. Informally, this can be thought of as grouping points that are “similar” in both P_src and P_tgt and then, for the explanation, showing the differences between the P_src members and P_tgt members of each group. Thus, domain knowledge is only necessary when choosing $k$ and when interpreting the *explanations* themselves--which is necessary for any explanation method.
>
> >  Looking at the k-mean explanation only on the specific evaluation shown in Table 1, it is possible to understand the shift quite well without resorting to the authors' k-OT approach…how to pick between $k$-$\mu$-Ex and $k$-$OT$-Ex)
>
> We have realized Table 1. was confusing and have updated it in the new version of the paper (apologies for the confusion). In the new version, you’ll see the baseline approach is the vanilla mean shift explanation (left of the dashed line) which operates on all 30K features. Although this could easily be truncated to the first $k$ features (similar to what is seen in $k$-sparse shift), this is difficult to determine where the truncation should happen without knowing the PercentExplained--which is provided in our two approaches ($k$-$\mu$-Ex and $k$-$OT$-Ex).
>
> As for the qualitative difference between $k$-$\mu$-Ex and $k$-$OT$-Ex, it is clear that $k$-$OT$-Ex explains more of the shift with smaller values of $k$, however, this is performed in a complicated mapping between P_src and P_tgt (since $k$-$OT$-Ex transports each sample uniquely as opposed to $k$-$\mu$-Ex which simply adds a constant to all samples). So, if a practitioner requires a higher fidelity solution at the cost of interpretability, they should use $k$-$OT$-Ex. Conversely, if they want a simpler (i.e. more naturally interpretable) solution and can tolerate it being less accurate, they should use $k$-$\mu$-Ex. Of course, it is possible to use both methods as well (our code is open-source and optimized for both approaches).
>
> > How would the CivilComments experiment compare against Shapley values
>
> While we compared against Shapley values as a baseline in the Adult Income experiment (and showed how this is less informative than our method), a comparison to Shapley values would be more difficult for this problem due to the dimensionality of the CivilComments experiment (30K features). While we could truncate the feature set to a smaller vocab count (e.g., the $k$ most common words), train a domain classifier, and explain that using Shapley values, if we wanted to adjust $k$ we would have to repeat the entire process. Unlike in our approach where adjusting $k$ is easy and requires minimal additional computation (only recalculating the PercentExplained).

---

> > ### Author Response · Authors · 2022-11-28
> > **Have we addressed your questions?**
> >
> > Hi Reviewer Lnku, thank you again for your helpful feedback. Have we adequately addressed your questions and concerns?
> >
> > In addition to reinforcing the ubiquity and importance of gaining a better understanding of distribution shifts, we hope to have shown that the level of required domain expertise is not much more than what is required for classic ML explanation methods (e.g., LIME). Additionally, we hope to have shown that the evaluations of our method are extensive (5 experiments on real-world data + 5 experiments on simulated data) given the context that this is a work in a new field (explaining distribution shifts) which has no formal benchmarks to evaluate against (on top of this being a general method in explainability, which is already hard to define and evaluate).
> >
> > Please let us know if we can answer any more questions. Thank you!

---

> > > ### Author Response · Authors · 2022-12-09
> > > **Author Response Follow-Up**
> > >
> > > Hi Reviewer Lnku, given our response detailing the human knowledge required (similar to that of typical ML interpretability methods) and detailing our evaluations (in the context of a new field), would you consider raising your score?
> > >
> > > Thank you again for your review and your time.

---

### Author Response · Authors · 2022-11-17
**General Response to Reviewers**

Dear reviewers,

We thank all reviewers for their helpful comments and feedback for improving the paper! We would like to highlight several important updates and clarifications for our paper before providing more detailed responses to each reviewer.

**Novelty of the problem**:  All reviewers are in agreement this is an important and novel area (“important research problem” - R-LnKU, “important and practical” - R-601f, “can help an operator investigate a distribution shift on real-world examples” - R-Sirc, “important problem” - R-xS74). However, our work does not address all issues in explaining distribution shifts. We heartily acknowledge there are limitations (as enumerated in Section 7: Limitations and in Appendix B), however, we urge the reviewers to reconsider these weaknesses from the perspective that this is an entirely new (yet, as we all agreed, a very important) area of research. No published baselines for shift explanations exist. So, instead, we have to compare against what we imagine could be a good baseline method (e.g., mean shift, explaining a domain classifier, etc.) and define the explainability metrics ourselves (which is already a difficult unsolved problem in explainability research [1], [2]). As this new area matures, there likely will be methods involving better interpretable mappings, interpretability metrics that are designed to better fit a specific context, or altogether different methods. However, one of our main goals is to introduce the problem of explaining distribution shifts and take the first step toward this task.  Overall, we hope these limitations of a novel research area are not rejection-worthy concerns as we heartily acknowledge the current limitations and hope future work will be able to expand and refine the problem and solutions.

**General updates to the paper**:
We have updated our overview figure (Figure 1.) to make the different types of mappings and their application areas more clear.
We added to our distribution shift explanation definition that, similar to ML model interpretability [1], a distribution shift explanation mapping can either be one that is intrinsically interpretable (see subsection 3.1) or a mapping which is explained via post-hoc explanation methods such as sets of input-output pairs (see subsection 6.2).
We have updated our table for the CivilComments experiment, caption, and have added an unrestricted mean shift explanation as a baseline (the original table had no baseline, since both $k-\mu-$Ex and $k-$OT$-$Ex are variants of $k$-sparse-shifts). We hope this clarifies how our method provides more information (both the word difference and how much that word explains the shift) and is more interpretable than a vanilla mean-shift explanation.


[1] Molnar, Christoph. Interpretable machine learning. Lulu. com, 2020.

[2] Lipton, Zachary C. "The mythos of model interpretability: In machine learning, the concept of interpretability is both important and slippery." Queue 16.3 (2018): 31-57.

---

### Decision · Program_Chairs · 2023-01-20

**Decision:**

Reject

**Justification For Why Not Higher Score:**

Overall structure, clarity and cohesion needs improvement.

**Justification For Why Not Lower Score:**

N/A

**Metareview: Summary, Strengths And Weaknesses:**

In this paper, authors present interpretable mappings to explain distribution shift. This is an important problem and a better understanding of the problem can be very helpful for the community. The paper is well-written and easy to follow. However, there are a few weaknesses that contributed to an agreement among reviewers to reject the paper. In our discussions, reviewer 6o1f nicely summarized them as follows:

- Overall structure and cohesion: The paper feels like a patchwork of 3-4 separate methods that targets different problem settings and applications, and does not have a cohesive motivation. As such, each method is not developed or motivated very thoroughly, and each method is only evaluated on 1-2 real datasets. For example, the authors introduce the  method in Section 6.1, but do not show results for it in the main paper at all -- only for one semi-synthetic dataset in the appendix. Similarly, the authors introduce the I2I counterfactual method in Section 6.2, which is compelling and well-evaluated, but has little connection to the k-sparse mappings discussed in the earlier parts of the paper.

- Lack of concrete metrics for evaluation: For most of the explanations proposed, the authors do not have a concrete way of evaluating its quality. The authors do propose the PercentExplained metric for the k-sparse transport and k-cluster transport methods. However, there are several issues with this metric which result in inflated values for the proposed methods over the baseline, which I point out in my review, and which the authors partly admit to. Short of a concrete metric, the only way to evaluate the quality of the explanation is through a visual inspection, which is imprecise at best.

- Lack of actionable insights for practitioners: While it is not expected for the authors to solve the entire distribution shift problem, the authors should at least demonstrate the utility of their explanations for real-world decision making. At a minimum, the authors should provide one or two experimental examples where practitioners use insights from the explanations to improve their model and mitigate the impact of the distribution shift (e.g. through additional data collection), perhaps in a semi-synthetic setting. Without these actionable insights, the utility of the explanation is less clear.

**Summary Of Ac-Reviewer Meeting:**

The main contributions of the paper and its strengths and weaknesses were discussed.

6o1f: There are several weaknesses that can be improved:

1) Clarity and structure of the paper
2) Lack of a metric for evaluation
3) Lack of actionable insights for practitioners

Lnku: Rebuttal has addressed some of the concerns but it is not clear how the results can be used in any way or what is the main insight.

xS74: After reading the others’ comments and the authors’ response, I lean towards rejection since I think the explaining methods also needs to be actionable, to be helpful for ML engineerings to adopt in practice